# UOTIP: Unbalanced Optimal Transport Map for Unpaired Inverse Problems

**Donggyu Lee** [* 1]    **Taekyung Lee** [* 2]    **Jaewoong Choi** [3]

## Abstract

We investigate unpaired image inverse problems, a challenging setting where only independent, non-paired sets of noisy measurements and clean target signals are available for training. We propose a novel inverse problem solver based on Unbalanced Optimal Transport, called **Unbalanced Optimal Transport Map for Inverse Problems (UOTIP)**. Our method formulates the reconstruction task—predicting clean target signals from noisy measurements—as learning a UOT Map from noisy measurement distribution to clean signal distribution by incorporating a likelihood-based cost function. By relaxing the exact marginal constraint, the UOT framework provides key advantages to our model: robustness to multi-level observation noise, adaptability to class imbalance between noisy and clean datasets, and generalizability to diverse noise-type scenarios. Furthermore, we theoretically demonstrate that incorporating a quadratic cost term ensures the existence and uniqueness of the transport map by satisfying the twist condition, even for ill-posed inverse problems. Our experiments demonstrate that UOTIP achieves state-of-the-art performance on unpaired image inverse problem benchmarks, across linear and nonlinear inverse problems.

## 1. Introduction

The **inverse problem** refers to the problem of recovering an unknown signal $\mathbf{x}$ from incomplete or noisy measurements $\mathbf{y}$ (Qayyum et al., 2022; Daras et al., 2024). This mathematical framework is fundamental to numerous scientific and engineering fields, including seismic imaging (Virieux & Operto, 2009), weather prediction (Huang et al., 2005), audio signal processing (Lemercier et al., 2024), and medical

---
[*]Equal contribution [1]Department of Mathematical Sciences, Seoul National University [2]AX/PI Center, Samsung Electronics [3]Sungkyunkwan University. Correspondence to: Jaewoong Choi <jaewoongchoi@skku.edu>.

*Proceedings of the 43rd International Conference on Machine Learning*, Seoul, South Korea. PMLR 306, 2026. Copyright 2026 by the author(s).

imaging (Song et al., 2021). In particular, this includes various *image reconstruction tasks*, such as image denoising (Zhang et al., 2017) and super-resolution (Tang et al., 2024; Dong et al., 2014). Formally, for $\mathbf{x} \in \mathbb{R}^n$ and $\mathbf{y} \in \mathbb{R}^m$, the inverse problem can be expressed as follows:

$$\mathbf{y} = \mathcal{A}(\mathbf{x}) + \mathbf{n}. \tag{1}$$

where $\mathcal{A}$ denotes a (possibly nonlinear) corruption operator and $\mathbf{n}$ represents measurement noise, which is typically assumed to be Gaussian $\mathcal{N}(0, \sigma_{\mathbf{y}}^2 I_m)$.

A key challenge in solving inverse problems is their *ill-posedness*, where the mapping $\mathbf{y} \mapsto \mathbf{x}$ may not have a unique solution (Engl & Ramlau, 2015). To address this, a Bayesian approach incorporates prior knowledge about $\mathbf{x}$ through the prior distribution $p(\mathbf{x})$, leading to a *Maximum a Posteriori* (MAP) estimate. Given a measurement $\mathbf{y}_0$, the MAP estimate is

$$\begin{aligned}
\hat{\mathbf{x}}_{\mathrm{MAP}}(\mathbf{y}_0) &= \arg\max_{\mathbf{x}} p(\mathbf{x}|\mathbf{y_0}) \\
&= \arg\min_{\mathbf{x}} \left[ -\log p(\mathbf{y}_0|\mathbf{x}) - \log p(\mathbf{x}) \right].
\end{aligned} \tag{2}$$

where $\log p(\mathbf{y}_0|\mathbf{x})$ represents the log-likelihood of the measurement $\mathbf{y}_0$ given the estimate $\mathbf{x}$. Intuitively, the MAP estimate addresses the ill-posedness of the inverse problem by selecting plausible samples $\mathbf{x}$ from the prior distribution $p(\mathbf{x})$ that also achieve a high likelihood for producing the observation $\mathbf{y}_0$. In this regard, selecting an appropriate prior distribution $p(\mathbf{x})$ for the target (clean) signal $\mathbf{x}$ is important.

Early approaches relied on hand-crafted priors, such as sparsity (Candès & Wakin, 2008), low-rank (Fazel et al., 2008), and total variation (Candès et al., 2006). However, such hand-crafted priors often fail to effectively characterize natural (desired) signals from unnatural signals (Ulyanov et al., 2018). To overcome this limitation, generative models have emerged as powerful alternatives for representing complex natural signals, such as GANs (Wang et al., 2022; Bora et al., 2017), VAEs (Goh et al., 2019), Optimal Transport Map (Tang et al., 2024; Korotin et al., 2023), and Diffusion models (Song et al., 2021; Chung et al., 2022).

We propose a novel approach that implicitly represents the prior distribution via Unbalanced Optimal Transport (UOT), a generalization of standard Optimal Transport. We refer

to our model as the ***Unbalanced Optimal Transport map for Inverse Problems (UOTIP)***. Specifically, we formulate the unpaired inverse problems as learning the UOT Map from the noisy measurement distribution to the target signal distribution. Our experiments demonstrate that our model achieves state-of-the-art performance among OT-based direct transport methods on unpaired inverse problem benchmarks across linear and nonlinear inverse problems. Moreover, by relaxing the exact marginal-matching constraint through UOT formulation, our model attains several critical advantages: **robustness to multi-level observation noise**, **adaptability to class imbalance** between unpaired datasets, and **generalizability to diverse noise-type scenarios**. These properties make our model particularly effective in real-world settings, where strict alignment between noisy observation and clean signal data is rarely achievable. Our contributions can be summarized as follows:

- We introduce the first model for unpaired inverse problems based on the Unbalanced Optimal Transport by introducing the likelihood cost function.

- We theoretically prove that incorporating a quadratic cost term ensures the existence and uniqueness of the OT inverse problem solver even for ill-posed inverse problems.

- Our model demonstrates state-of-the-art performance among OT-based direct transport methods on both linear and nonlinear inverse problem benchmarks.

- Our UOT formulation offers our model robustness to multi-level observation noise and adaptability to class imbalance, making it effective in real-world scenarios.

## 2. Background

**Notations and Assumptions** Let $\mathcal{X}$, $\mathcal{Y}$ be two compact complete metric spaces, and let $\mu$ and $\nu$ denote probability distributions on $\mathcal{Y}$ and $\mathcal{X}$, respectively. Both $\mu$ and $\nu$ are assumed to be absolutely continuous with respect to the Lebesgue measure. Throughout this work, the source distribution $\mu$ and the target distribution $\nu$ represent the distributions of noisy measurements $\mathbf{y}$ and clean target signals $\mathbf{x}$, respectively. For a measurable map $T$, $T_{\#}\mu$ represents the pushforward distribution of $\mu$. The set $\Pi(\mu, \nu)$ denote the set of joint probability distributions on $\mathcal{Y} \times \mathcal{X}$ whose marginals are $\mu$ and $\nu$, respectively. Finally, given a function $f : \mathbb{R} \to [-\infty, \infty]$, its convex conjugate $f^*$ is defined as $f^*(y) = \sup_{x \in \mathbb{R}}\{\langle x, y \rangle - f(x)\}$.

**Optimal Transport** The *Optimal Transport (OT)* problem seeks a cost-minimizing way to transport one probability distribution to another (Villani et al., 2009; Santambrogio, 2015). Formally, the OT problem aims to map a source distribution $\mu \in \mathcal{P}(\mathcal{Y})$ to a target distribution $\nu \in \mathcal{P}(\mathcal{X})$ while minimizing a given cost function $c(\cdot, \cdot)$. The Monge's OT

problem (Eq. 3) involves finding a deterministic transport map $T$ such that $T_{\#}\mu = \nu$ (Monge, 1781) (Fig. 1).

$$C(\mu, \nu) := \inf_{T_{\#}\mu = \nu}\left[\int_{\mathcal{Y}} c(\mathbf{y}, T(\mathbf{y}))d\mu(\mathbf{y})\right]. \quad (3)$$

Intuitively, Monge's OT problem seeks an optimal transport map $T^\star$ that generates the target distribution $\nu$ by mapping each input $\mathbf{y}$ in a way that minimizes the transport cost $c(\mathbf{y}, T(\mathbf{y}))$. However, Monge's formulation is non-convex, and a deterministic optimal transport map $T^\star$ may not exist depending on the properties of $\mu$ and $\nu$ (Villani et al., 2009; Choi et al., 2025b). To address these limitations, Kantorovich proposed a relaxed formulation of the OT problem (Kantorovich, 1948), which models transport via stochastic couplings $\pi$:

$$C_{ot}(\mu, \nu) := \inf_{\pi \in \Pi(\mu, \nu)}\left[\int_{\mathcal{Y} \times \mathcal{X}} c(\mathbf{y}, \mathbf{x})d\pi(\mathbf{y}, \mathbf{x})\right]. \quad (4)$$

Here, $\pi \in \Pi(\mu, \nu)$ denotes a coupling of $\mu$ and $\nu$, i.e., a joint distribution with marginals $\mu$ and $\nu$. Unlike Monge's OT problem, the Kantorovich formulation (Eq. 4) guarantees the existence of an optimal transport plan $\pi^\star$ under mild assumptions (Villani et al., 2009). Furthermore, if $\mu$ and $\nu$ are absolutely continuous with respect to the Lebesgue measure (our assumption), the optimal transport map $T^*$ exists and the optimal plan is induced by it, i.e., $\pi^\star = (Id \times T^\star)_{\#}\mu$ (Villani et al., 2009).

The goal of ***Neural Optimal Transport (Neural OT)*** is to learn the optimal transport map $T^\star$ using neural networks. Rout et al. (2022); Fan et al. (2023) proposed a method based on the semi-dual formulation of OT (Eq. 5), where the potential function $v_\phi$ and the transport map $T_\theta$ are both parameterized by neural networks. Intuitively, analogous to GANs, the potential function $v_\phi$ and the transport map $T_\theta$ play a similar role to the discriminator and generator.

$$\mathcal{L}_{v_\phi, T_\theta} = \sup_{v_\phi}\left[\int_{\mathcal{Y}} \inf_{T_\theta}\left[c\left(\mathbf{y}, T_\theta(\mathbf{y})\right) - v_\phi\left(T_\theta(\mathbf{y})\right)\right]d\mu(\mathbf{y}) \right.$$
$$\left. + \int_{\mathcal{X}} v_\phi(\mathbf{x})d\nu(\mathbf{x})\right]. \quad (5)$$

**Unbalanced Optimal Transport** The *Unbalanced Optimal Transport (UOT)* problem (Chizat et al., 2018; Liero et al., 2018) is a generalization of the standard OT problem (Eq. 4) that allows flexibility in the source distribution $\mu$ and target distribution $\nu$:

$$C_{uot}(\mu, \nu) = \inf_{\pi \in \mathcal{M}_+(\mathcal{Y} \times \mathcal{X})}\left[\int_{\mathcal{Y} \times \mathcal{X}} c(\mathbf{y}, \mathbf{x})d\pi(\mathbf{y}, \mathbf{x}) \right.$$
$$\left. + D_{\Psi_1}(\pi_0 \| \mu) + D_{\Psi_2}(\pi_1 \| \nu)\right], \quad (6)$$

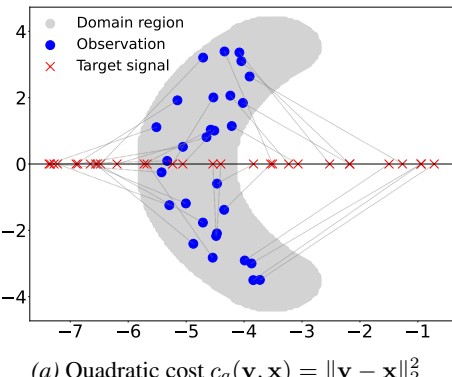

*(a)* Quadratic cost $c_q(\mathbf{y}, \mathbf{x}) = \|\mathbf{y} - \mathbf{x}\|_2^2$

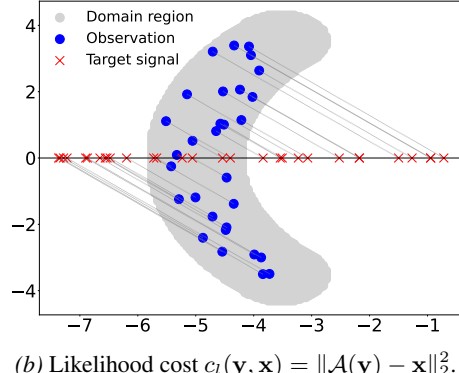

*(b)* Likelihood cost $c_l(\mathbf{y}, \mathbf{x}) = \|\mathcal{A}(\mathbf{y}) - \mathbf{x}\|_2^2$.

*Figure 1.* **OT Maps under different cost functions** ($\mathcal{A}$: projection onto $x$-axis along $\mathbf{d} = (1, -1)$)

where $\mathcal{M}_+(\mathcal{Y} \times \mathcal{X})$ denotes the set of positive Radon measures on $\mathcal{Y} \times \mathcal{X}$. The terms $D_{\Psi_1}$ and $D_{\Psi_2}$ are $f$-divergences induced by convex functions $\Psi_i$, penalizing deviations of the marginals of $\pi$ from $\mu$ and $\nu$, respectively, i.e., $D_{\Psi_i}(\pi_j \| \eta) = \int \Psi_i \left( \frac{d\pi_j(x)}{d\eta(x)} \right) d\eta(x)$. Note that the standard OT enforces exact matching of the marginals, i.e., $\pi_0 = \mu, \pi_1 = \nu$. In contrast, UOT relaxes this constraint by adding $f$-divergence penalties on the marginals. By minimizing these penalty terms, the marginals of the optimal UOT plan $\pi_u^\star$ are softly matched to $\mu$ and $\nu$, i.e., $\pi_{u,0}^\star \approx \mu$ and $\pi_{u,1}^\star \approx \nu$.

This relaxation provides UOT robustness to outliers (Séjourné et al., 2022; Gazdieva et al., 2025) and the ability to handle class imbalance between marginal distributions (Eyring et al., 2024; Lee et al., 2025). Here, the class imbalance problem refers to the scenario where the source and target distributions consist of different proportions of classes. The UOT map can be interpreted as the OT Map between rescaled marginal distributions $\pi_{u,0}^\star(\mathbf{y}) = \Psi_1^{*\prime}(-v^{\star c}(\mathbf{y})[1].)\mu(\mathbf{y})$ and $\pi_{u,1}^\star(\mathbf{x}) = \Psi_2^{*\prime}(-v^\star(\mathbf{x}))\nu(\mathbf{x})$, where $v^\star$ denotes the optimal potential (Eq. 10) (Choi et al., 2023). These rescaling factors $\Psi_i^{*\prime}$ enables the UOT Map $T_u^\star$ to naturally handle class imbalance by reweighting source samples. For example, $T_u^\star$ can make a correspondence between a mode representing 20% of the source distribution and a mode representing 30% of the target distribution.

Choi et al. (2023) introduced a Neural OT model for the UOT problem with the quadratic cost $c(\mathbf{y}, \mathbf{x}) = \|\mathbf{y} - \mathbf{x}\|_2^2$, called *UOTM*, and applied this to generative modeling. We extend the Neural UOT model to inverse problems by generalizing the cost function. In Sec 3, we demonstrate how inverse problems can be naturally formulated within the Neural OT framework and present our model.

---

[1]where $v^c(\mathbf{y})$ denotes the $c$-transform of $v$, which is defined as $v^c(\mathbf{y}) = \inf_{\mathbf{x} \in \mathcal{X}} (c(\mathbf{y}, \mathbf{x}) - v(\mathbf{x}))$

## 3. Method

In this section, we present our UOT-based model for inverse problems, called ***Unbalanced Optimal Transport Map for Inverse Problems (UOTIP)***. The core idea of UOTIP is to formulate the inverse problem solver as an Unbalanced Optimal Transport (UOT) map from the noisy measurements $\mathbf{y}$ to the target signal $\mathbf{x}$ (Eq. 1). In Sec 3.1, we introduce our UOT-based formulation of inverse problems. In Sec 3.2, we describe the learning objective, which is derived as a generalization of the vanilla Neural UOT model (Choi et al., 2023) under the proposed cost function.

### 3.1. Optimal Transport Formulation of Unpaired Inverse Problems

**Task Formulation** In this subsection, we interpret the inverse problems solver as a Neural (Unbalanced) Optimal Transport Map $T^\star$ that maps the noisy measurements $\mathbf{y}$ to the target signal $\mathbf{x}$ under an appropriate cost function $c(\cdot, \cdot)$ (to be specified in Eq. 9).

Our main target task is the **unpaired image inverse problems with a known corruption operator $\mathcal{A}$ and unknown noise level $\sigma_y > 0$ (Eq. 1)**. (The case of an unknown corruption operator will be discussed in Sec 4.3.) Formally, let $Y = \{\mathbf{y}_i : \mathbf{y}_i \in \mathcal{Y}, i = 1, \cdots, M\}$ and $X = \{\mathbf{x}_j : \mathbf{x}_j \in \mathcal{X}, j = 1, \cdots, N\}$ denote the sets of *noisy measurements* and *target signals (clean image)*, respectively. Under the unpaired setting, $Y$ and $X$ are independently sampled from the source distribution $\mu$ and the target distribution $\nu$. Then, our goal is to learn an inverse problem solver $T$:

$$T : \mathcal{Y} \to \mathcal{X}, \qquad \mathbf{y} \mapsto T(\mathbf{y}) \qquad (7)$$

from these unpaired training data.

To align $T$ with the MAP estimate principles, we seek a solver that simultaneously reflects the target prior and the observation likelihood:

(i) The outputs of a solver should be consistent with the target signal distribution, ideally satisfying $T_{\#}\mu = \nu$.

(ii) For any observation $\mathbf{y}_0$, the estimate $T(\mathbf{y}_0)$ should maximize the log-likelihood $\log p(\mathbf{y}_0|\cdot)$.

Here, the first condition (i) ensures that the predicted target signal $T(\mathbf{y})$ is physically plausible (prior fidelity). The condition (ii) ensures the reconstruction remains faithful to the specific measurement (data fidelity).

Our main observation is that **these two conditions can be naturally interpreted through the (Unbalanced) Optimal Transport** (Eq. 3 and 6). Formally, in the OT problem, the OT map $T^\star$ is defined as the transport cost minimizer over valid transport maps:

(a) Valid transport maps satisfying $T^\star_\# \mu = \nu$.

(b) Each $\mathbf{y}$ is mapped to minimize the (overall) transport cost $\int_{\mathcal{Y}} c(\mathbf{y}, T^\star(\mathbf{y})) d\mu(\mathbf{y})$.

Therefore, by construction, $T^\star$ satisfies the prior fidelity in condition (i). **Our approach is to design an appropriate cost function $c(\cdot, \cdot)$ so that the second condition (ii) is also satisfied**. Under this formulation, the Neural OT framework effectively serves as a global, unpaired MAP estimator.

**Likelihood Cost and MAP Estimate**    Our goal is to solve inverse problems using a Neural OT model. To adapt the general OT map as an inverse problem solver, we introduce the *likelihood cost function* that can be tailored to each inverse problem.

$$c_l(\mathbf{y}, \mathbf{x}) = \|\mathcal{A}(\mathbf{x}) - \mathbf{y}\|_2^2 \qquad (8)$$

Note that under the assumption of Gaussian measurement noise, $c_l$ is proportional to the negative log-likelihood of the observation $-\log p(\mathbf{y} \mid \mathbf{x})$. Fig. 1 illustrates how the OT map $T^\star$ changes under the likelihood cost and the standard quadratic cost. While derived from Gaussian assumptions, we experimentally demonstrate that this cost function generalizes effectively to other noise distributions, such as Laplace and Poisson noise (Sec 4.2).

Interestingly, the Neural OT model with the likelihood cost admits a **MAP interpretation**. In particular, adopting the cost function $c(\mathbf{y}, \mathbf{x}) = -\log p(\mathbf{y}|\mathbf{x})$ within the OT framework is equivalent to minimizing the negative log-posterior $-\log p(\mathbf{x}|\mathbf{y})$, :

$$
\begin{aligned}
C_{ot}(\mu, \nu) &= \inf_{\pi \in \Pi(\mu,\nu)} \left[ \int_{\mathcal{Y} \times \mathcal{X}} -\log p(\mathbf{x}|\mathbf{y}) d\pi(\mathbf{y}, \mathbf{x}) \right] \\
&= \inf_{\pi \in \Pi(\mu,\nu)} \Big[ \int_{\mathcal{Y} \times \mathcal{X}} -\log p(\mathbf{y}|\mathbf{x}) - \log p(\mathbf{x}) \\
&\qquad\qquad + \log p(\mathbf{y}) d\pi(\mathbf{y}, \mathbf{x}) \Big] \\
&= \inf_{\pi \in \Pi(\mu,\nu)} \left[ \int_{\mathcal{Y} \times \mathcal{X}} -\log p(\mathbf{y}|\mathbf{x}) d\pi(\mathbf{y}, \mathbf{x}) \right]
\end{aligned}
$$

The last equality holds because, for $\pi \in \Pi(\mu, \nu)$, the

marginal distributions are fixed to $\pi_0 = \mu$ and $\pi_1 = \nu$. Consequently, **the OT formulation implicitly incorporates the target signal prior $p(\mathbf{x})$ through the distributional constraints of the valid transport map.**

**Well-posedness via Quadratic Cost**    To complement the likelihood cost $c_l$, we incorporate the quadratic cost $c_q$. The resulting overall cost function is expressed as follows:

$$c(\mathbf{y}, \mathbf{x}) = \tau \big( c_l(\mathbf{y}, \mathbf{x}) + c_q(\mathbf{y}, \mathbf{x}) \big) \qquad (9)$$

where $c_l(\mathbf{y}, \mathbf{x}) = \|\mathcal{A}(\mathbf{x}) - \mathbf{y}\|_2^2$ and $c_q(\mathbf{y}, \mathbf{x}) = \|\mathbf{y} - \mathbf{x}\|_2^2$. Here, $\tau$ denotes the cost intensity parameter.

This additional quadratic term serves two purposes. First, it establishes the theoretical well-posedness of the OT inverse problem solver. Inverse problems are typically ill-posed. From the OT perspective, this ill-posedness can cause the likelihood cost $c_l$ to violate the *twist condition*, which is critical for guaranteeing the existence and uniqueness of the OT Map $T^\star$. Under mild assumptions, incorporating the quadratic cost $c_q$ ensures that the overall cost function $c(\cdot, \cdot)$ satisfies the twist condition, thereby guaranteeing the **existence and uniqueness of the OT inverse problem solver** (See Appendix A for details.)[2].

**Proposition 3.1.** *For ill-posed inverse problems such as Gaussian deblurring or HDR reconstruction, the additional quadratic cost $c_q$ ensures that the overall cost function $c(\cdot, \cdot)$ (Eq. 9) satisfies the twist condition, which guarantees the existence and uniqueness of the OT map. Formally, if $\mathcal{A}$ is L-Lipschitz continuous, the cost function $c_l(\mathbf{y}, \mathbf{x}) + \lambda c_q(\mathbf{y}, \mathbf{x})$ satisfies the twist condition when $\lambda > L$.*

Second, from a practical perspective, this quadratic cost allows us to extend our model to scenarios with an unknown corruption operator (by using quadratic cost only). As shown in the ablation study in Sec 4.3, our quadratic-only variant still achieves competitive performance on linear inverse problems even without explicit knowledge of the corruption operator $\mathcal{A}$.

### 3.2. Proposed Model

In this subsection, we present the learning objective of our UOTIP model and then highlight the theoretical advantages of adopting the UOT map over the standard OT map for inverse problems.

**Learning Objective**    Our approach is to learn the UOT Map $T^\star_u$ from the noisy measurement distribution $\mu$ to the target signal distribution $\nu$ using a neural network $T_\theta$. To this end, we employ the UOTM framework (Choi et al.,

---

[2]In practice, we additionally require the forward operator $\mathcal{A}$ to be differentiable because the gradient of $\mathcal{A}$ is required in the optimization process (Algorithm 1).

*Table 1.* **Quantitative results on unpaired inverse problems**: two linear (Gaussian deblurring, Super-resolution) and two nonlinear (HDR Reconstruction, Nonlinear Deblurring). The **boldface** and underlined values indicate the best and second-best performance. Our model consistently achieves strong performance, attaining the best scores in nearly all cases.

| Task | Method | FFHQ | | | AFHQ | | |
|---|---|---|---|---|---|---|---|
| | | PSNR (↑) | SSIM (↑) | FID (↓) | PSNR (↑) | SSIM (↑) | FID (↓) |
| Gaussian Deblurring | NOT (Korotin et al., 2023) | 20.11 | 0.6035 | 52.901 | 19.99 | 0.5472 | 58.927 |
| | OTUR (Wang et al., 2022) | 23.82 | 0.7106 | 24.337 | 23.91 | 0.6777 | 30.773 |
| | RCOT (Tang et al., 2024) | 22.07 | 0.5492 | 123.692 | 22.34 | 0.5365 | 132.465 |
| | UOTIP (Ours) | **24.06** | **0.7139** | **21.210** | **24.22** | **0.6804** | **12.566** |
| Super-resolution 4× | NOT (Korotin et al., 2023) | 20.13 | 0.6257 | 50.066 | 20.14 | 0.5833 | 44.252 |
| | OTUR (Wang et al., 2022) | 24.09 | 0.7243 | 22.751 | 24.71 | 0.7079 | 19.575 |
| | RCOT (Tang et al., 2024) | 24.05 | 0.6820 | 118.776 | **25.04** | 0.7137 | 69.072 |
| | UOTIP (Ours) | **24.35** | **0.7371** | 19.475 | 24.97 | **0.7142** | **15.939** |
| HDR Reconstruction | NOT (Korotin et al., 2023) | 21.24 | 0.7978 | 25.842 | 23.36 | 0.8179 | 10.528 |
| | OTUR (Wang et al., 2022) | 25.32 | 0.8545 | **16.458** | 26.25 | 0.8542 | **7.227** |
| | RCOT (Tang et al., 2024) | 19.26 | 0.6755 | 33.422 | 18.99 | 0.7060 | 27.767 |
| | UOTIP (Ours) | **26.02** | **0.8642** | 20.840 | **26.40** | **0.8653** | 7.897 |
| Nonlinear Deblurring | NOT (Korotin et al., 2023) | 21.37 | 0.7373 | 43.661 | 23.03 | 0.7271 | 17.377 |
| | OTUR (Wang et al., 2022) | 26.94 | 0.8594 | 12.538 | 26.09 | 0.8253 | 7.651 |
| | RCOT (Tang et al., 2024) | 25.14 | 0.7221 | 52.268 | 24.48 | 0.7172 | 29.902 |
| | UOTIP (Ours) | **28.52** | **0.8841** | **11.370** | **27.74** | **0.8589** | **5.113** |

2023), which is based on the semi-dual formulation of the UOT (Vacher & Vialard, 2023) (Eq. 10).

$$C_{uot}(\mu, \nu) = \sup_{v \in \mathcal{C}} \left[ \int_{\mathcal{Y}} -\Psi_1^* \left( -v^c(\mathbf{y}) \right) d\mu(\mathbf{y}) \right.$$
$$\left. + \int_{\mathcal{X}} -\Psi_2^* (-v(\mathbf{x})) d\nu(\mathbf{x}) \right], \quad (10)$$

where $v^c(\mathbf{y})$ denotes the $c$-transform of $v$, which is defined as $v^c(\mathbf{y}) = \inf_{\mathbf{x} \in \mathcal{X}} (c(\mathbf{y}, \mathbf{x}) - v(\mathbf{x}))$. We refer to the $v \in \mathcal{C}$ as the potential function. Then, $T_\theta$ is parameterized to approximate the optimal UOT Map $T_u^\star$ as follows:

$$T_\theta(\mathbf{y}) \in \operatorname*{arginf}_{\mathbf{x} \in \mathcal{X}} \left[ c(\mathbf{y}, \mathbf{x}) - v(\mathbf{x}) \right]$$
$$\Leftrightarrow \quad v^c(\mathbf{y}) = c(\mathbf{y}, T_\theta(\mathbf{y})) - v(T_\theta(\mathbf{y})), \quad (11)$$

Note that this parameterization leverages the optimality condition, which is satisfied by the UOT Map $T_u^\star$ and the optimal potential function $v^\star$ (Choi et al., 2023). Finally, by substituting $v^c$ from Eq. 10 with Eq. 11 and parameterizing the potential function as $v_\phi$ with neural network, we obtain the following learning objective using the cost function from Eq. 9:

$$\mathcal{L}_{v_\phi, T_\theta}$$
$$= \inf_{v_\phi} \int_{\mathcal{Y}} \Psi_1^* \left( -\inf_{T_\theta} \left[ c(\mathbf{y}, T_\theta(\mathbf{y})) - v_\phi(T_\theta(\mathbf{y})) \right] \right) d\mu(\mathbf{y})$$
$$+ \int_{\mathcal{X}} \Psi_2^* (-v_\phi(\mathbf{x})) d\nu(\mathbf{x}). \quad (12)$$

Here, the convex conjugates $\Psi_1^*$ and $\Psi_2^*$ are monotone increasing convex functions, determined by the marginal penalty terms in the UOT problem (Eq. 6). For example, if

$D_{\Psi_i}$ is the KL divergence, then $\Psi_i^*(\cdot) = \exp(\cdot) - 1$. Moreover, our UOT map objective reduces to the standard OT map variant (Eq. 5) when $\Psi_i^*(\cdot) = Identity(\cdot)$, which corresponds to selecting $\Psi_i$ as the convex indicator function at $\{1\}$. Hence, Neural UOT serves as a generalization of the Neural OT. For the training algorithm, refer to Algorithm 1.

**Unbalanced OT vs. OT** Our goal is to solve inverse problems under an unpaired setting, where the training datasets $Y$ and $X$ are not given as paired samples. Relaxing marginal distribution constraints through UOT provides several distinctive advantages for the UOT map $T_u^\star$:

(a*) The relaxed constraints allow the map to attain higher-likelihood regions in terms of both prior and data fidelity.

(b*) UOT naturally handles class imbalance between the source and target distributions. This is particularly useful for multi-level measurement noise $\sigma_y$, where multiple noisy observations $\mathbf{y}$ may correspond to a single target signal $\mathbf{x}$.

(c*) UOT improves training dynamics in Neural OT frameworks by upper-bounding the gradient norm of the potential functions, leading to better target distribution fidelity. (Choi et al., 2024b).

These properties make the UOT map particularly well-suited for unpaired inverse problems. Specifically regarding (a*), the relaxed constraint enables $T_u^\star$ to satisfy a softer condition (i-1) (replacing (i) in Sec 3.1) by focusing more on high-density regions.

(i-1) $T(\mathbf{y})$ attains a high likelihood under the target signal distribution, i.e., $\nu(T(\mathbf{y})) > M$ for all $\mathbf{y}$ for some threshold $M > 0$.

For the inverse problems, condition (i-1) results in predicted signals that are more physically plausible. Furthermore, the UOT achieves a smaller transport cost compared to the OT by permitting small marginal mismatches, which enables more effective overall cost minimization.

$$\int c(\mathbf{y}, \mathbf{x}) \, d\pi_{ot}^{\star}(\mathbf{y}, \mathbf{x}) \geq \int c(\mathbf{y}, \mathbf{x}) \, d\pi_{uot}^{\star}(\mathbf{y}, \mathbf{x}). \quad (13)$$

For our likelihood cost, this means higher data fidelity, ensuring $\mathcal{A}(T(\mathbf{y})) \approx \mathbf{y}$. In conclusion, the Neural UOT framework achieves a superior global MAP estimate over standard Neural OT by simultaneously improving prior and data fidelity through marginal relaxation. Given these benefits, our model is built upon the UOT Map.

## 4. Experiments

In this section, we evaluate the performance of our model from diverse perspectives.

- In Sec 4.1, we assess our model on four inverse problem benchmarks, including linear inverse problems and nonlinear inverse problems.

- In Sec 4.2, we investigate several advantages of our model, derived from the Unbalanced Optimal Transport formulation, such as handling multi-noise level observations, addressing the class imbalance problem, and managing diverse noise types.

- In Sec 4.3, we conduct an ablation study on the likelihood cost function $c_l$ (See Appendix E.1 for additional results on the cost intensity hyperparameter $\tau$).

**Setting** We evaluated our model on four unpaired inverse problem benchmarks: *two linear (Gaussian deblurring, super-resolution)* and *two nonlinear (HDR reconstruction, nonlinear deblurring)*, with additive white Gaussian noise ($\sigma_{\mathbf{y}} = 0.05$). For the super-resolution, the quadratic cost $c_q(\mathbf{y}, \mathbf{x})$ cannot be directly applied because $\mathbf{y}$ and $\mathbf{x}$ have different dimensions. Hence, we introduce bicubic interpolation $Q$ on the lower-resolution image $\mathbf{y}$ and define the modified cost as $c_q(\mathbf{y}, \mathbf{x}) = \|Q(\mathbf{y}) - \mathbf{x}\|_2^2$.[3] Additional implementation details are provided in Appendix B.

We test our model on two image datasets: **FFHQ** (Karras et al., 2019) and **AFHQ-dog** (Choi et al., 2020), resized to $128 \times 128$. For quantitative evaluation, we employ PSNR and

SSIM for assessing pixel-level similarity with the ground-truth target signal, and FID for perceptual quality. We compare our model against existing approaches for unpaired inverse problems: GAN-based *OTUR (Wang et al., 2022)* and OT-map–based *NOT (Korotin et al., 2023)* and *RCOT (Tang et al., 2024)*).

### 4.1. Unpaired Inverse Problem Benchmarks

Table 1 shows the quantitative results on four inverse problems across two datasets. Our UOTIP achieves the best performance on almost all metrics across inverse problems and datasets. Specifically, in Gaussian deblurring and nonlinear deblurring, our method consistently outperforms all other approaches across all three metrics on both datasets. In super-resolution, our method achieves the best scores except for PSNR on AFHQ, where RCOT attains a comparable PSNR score to ours. Here, our approach attains significantly better FID scores on both datasets, indicating that our method attains superior fidelity of target signals compared to other methods.

Fig. 2 shows qualitative comparisons on the FFHQ dataset (see Figure 6 in the Appendix D.1 for AFHQ results and Appendix D.2 for additional examples). The results further demonstrate our superior image fidelity. NOT often overemphasize features, such as contours and textures, while RCOT fails to reliably remove degradations. OTUR effectively restores degradations but often distorts or over-smooths fine details, including eyes, mouths, and textural details. In contrast, our model produces sharper images with well-preserved textures.

### 4.2. Investigating Advantages from the (U)OT Framework

In this subsection, we examine the benefits of our UOT formulation for unpaired inverse problems. Specifically, our UOTIP is tested under ***multi-level noise observations, class imbalance settings,*** and ***various noise types***. The first two advantages stem from the unbalancedness of our UOT formulation. As discussed in Sec. 3.2, unlike the standard OT Map, the UOT Map $T_u^{\star}$ allows rescaling of each sample $T_u^{\star}(\mathbf{y})$. This flexibility provides robustness to handle varying noise levels and class imbalance in inverse problems. For the last advantage, we evaluate whether our model can generalize to various noise types. This robustness arises from the OT formulation itself.

**Multi-level Observation Noise** Most existing unpaired inverse problem solvers (Tang et al., 2024; Wang et al., 2022) assume a single fixed noise level $\sigma_{\mathbf{y}} > 0$. However, in realistic scenarios, **observation noise can vary across samples**. To test robustness under such conditions, we design experiments on the FFHQ dataset where noise is drawn

---

[3]Note that under this modified cost, the twist condition in Prop. 3.1 is no longer satisfied. Hence, the existence of the OT map is not theoretically guaranteed. Nevertheless, as shown in Table 1, UOTIP performs well on super-resolution in practice. We attribute this empirical success to the local smoothing inductive bias of the generator network architectures.

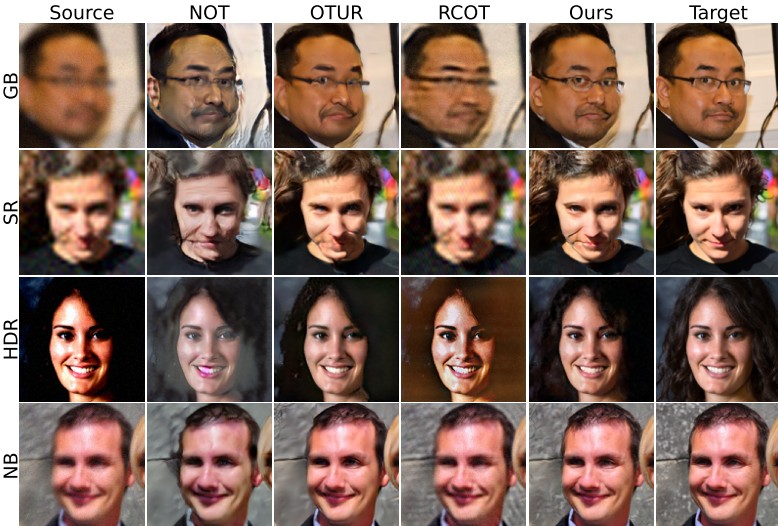

*Figure 2.* **Comparison of inverse problem solvers on FFHQ** for four tasks: Gaussian deblurring (GB), Super-resolution (SR), High dynamic range reconstruction (HDR), and Nonlinear deblurring (NB). Our model produces higher fidelity images with well-preserved textures.

*Table 2.* **Quantitative results under multi-level observation noise** for linear inverse problems on FFHQ. Our model exhibits superior robustness across noise levels.

| Method | Gaussian Deblurring | | | Super Resolution 4× | | |
|---|---|---|---|---|---|---|
| | PSNR (↑) | SSIM (↑) | FID (↓) | PSNR (↑) | SSIM (↑) | FID (↓) |
| NOT (Korotin et al., 2023) | 19.07 | 0.5491 | 98.558 | 18.91 | 0.5653 | 99.053 |
| OTUR (Wang et al., 2022) | 22.55 | 0.6417 | 67.323 | 23.20 | 0.6681 | 70.223 |
| OTIP (Ours-OT) | 22.87 | 0.6562 | 91.309 | 23.21 | 0.6732 | 85.674 |
| UOTIP (Ours) | **23.04** | **0.6649** | **65.664** | **23.30** | **0.6864** | **58.406** |

from a mixture of four levels: $\sigma_{\mathbf{y}} \in \{0.025, 0.05, 0.1, 0.2\}$, instead of fixing $\sigma_{\mathbf{y}} = 0.05$. For each level, degraded images are independently sampled in proportions 4:3:2:1, so that multiple noisy observations $\mathbf{y} \sim \mu$ may correspond to the same clean target (Fig. 5). This setup evaluates whether our UOT-based formulation can effectively handle heterogeneous noise distributions. We compare UOTIP against *OTUR* (best baseline in Sec 4.1), *NOT* (OT-map model), and an OT variant of our model (*Ours-OT*), obtained by setting $\Psi_1(\cdot) = \Psi_2(\cdot) = \text{Identity}(\cdot)$ in Eq. 12. Since robustness to multi-level noise arises only from the unbalanced formulation, for completeness, we also compare with the OT-variant of our model.

Table 2 presents results on linear inverse problems (see Table 5 in Appendix E.2 for nonlinear results and Appendix D.3 for qualitative examples). Our UOTIP achieves the best performance on all metrics. Specifically, our model outperforms OTUR (the strongest baseline in standard benchmarks) and the OT-variant of our model (OTIP). In particular, UOTIP shows a clear improvement in FID over OTIP, highlighting its superior perceptual fidelity.

**Class Imbalance** We further evaluate UOTIP under class imbalance, where **source and target distributions are multi-modal but differ in their mode proportions**. This situation can naturally occur in unpaired settings when measurements and clean signals are obtained from different data sources. For example, in super-resolution, large amounts of low-resolution data may be available, whereas high-resolution data may be limited. In this case, additional public high-resolution datasets may then be incorporated, leading class imbalance between modes. To examine this, we constructed datasets by combining AFHQ-cat with AFHQ-dog (Choi et al., 2020). For each imbalance ratio $k \in \{1, 2, 3, 4\}$, we construct a target dataset $X$ of 6,000 images by mixing cat and dog samples at a $k : 1$ (cat: dog) ratio. The source dataset $Y$ is fixed at a $1 : 1$ class ratio by downsampling cat images from $X$, and is then generated by applying the degradation/measurement operator to this subset.

Fig 3 presents the quantitative results of PSNR and FID metric. For clarity, we visualized only two metrics, because the two pixel-level measures (PSNR, SSIM) exhibited similar trends (See Table 6 Appendix E.2 for the complete table). Similar to the multi-level noise experiments, our UOT-based model consistently outperforms the baselines under class imbalance, achieving the best or at least comparable performance across all metrics. OTUR and Ours-OT achieve intermediate performance, whereas NOT mostly yields the

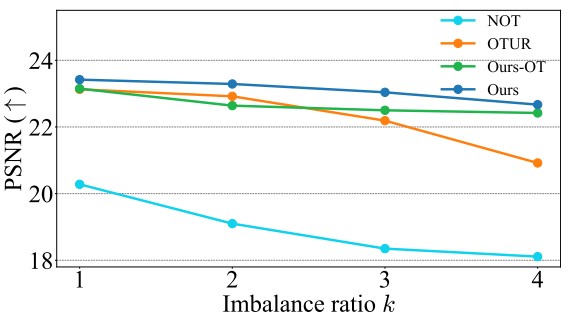 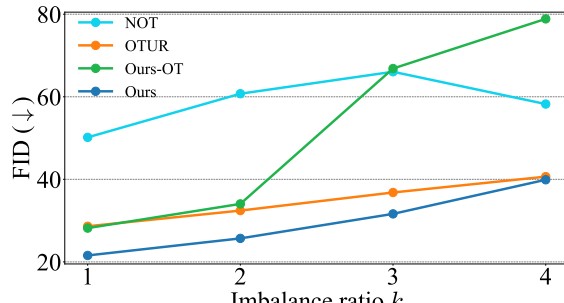

*Figure 3.* **Quantitative results under class imbalance** with imbalance ratio $k$ between AFHQ-cat and AFHQ-dog for the Gaussian deblurring (cat:dog $1 : 1 \rightarrow k : 1$). Our method achieves superior performance and demonstrates greater robustness across various ratios.

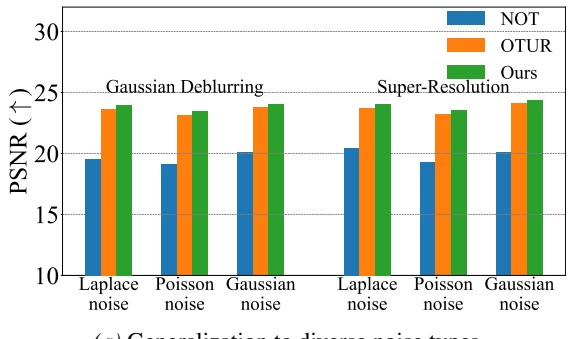 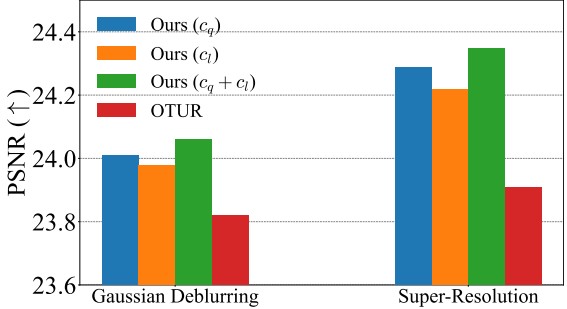

*(a)* Generalization to diverse noise types

*(b)* Ablation study on cost functions

*Figure 4.* **PSNR ($\uparrow$) results for diverse noise types and cost ablation.** Complete results on all three metrics are provided in Appendix E.2. In Fig.4a, our model generalizes well across different noise types. In Fig.4b, the full model achieves the best overall performance, and even the blind-corruption variant (using only $c_q$) outperforms the previous state-of-the-art OTUR model.

lowest results.

**Diverse Noise Types**    We incorporate the likelihood cost $c_l(\mathbf{y}, \mathbf{x})$, which is based on the negative log-likelihood of the Gaussian measurement noise, to guide the UOT map as our inverse problem solver. In this section, we examine the **robustness of our model under noise-type mismatch** by testing on alternative noise distributions while keeping the likelihood cost $c_l$ fixed as Gaussian. Intuitively, the UOT (or OT) map (Eq. 6) is defined as the transport cost minimizer (Condition (b)), i.e., enforcing $\mathcal{A}(T(\mathbf{y})) \approx \mathbf{y}$, among the valid transport maps (Condition (a)) in Sec 3.1. In this respect, minimizing $c_l$ is expected to yield robustness across noise types, since enforcing $\mathcal{A}(T(\mathbf{y})) \approx \mathbf{y}$ does not strictly depend on the Gaussian assumption. To validate this, we test our model under two alternative noise distributions: *Laplace noise* and *Poisson noise* (See Appendix B for details).

Fig. 4a presents PSNR results on linear inverse problems on the FFHQ dataset. Due to page constraints, only PSNR is shown here; complete results across all three metrics are provided in Table 7 of Appendix E.2. Our model attains strong performance across various noise types compared to other existing approaches: NOT and OTUR. These results demonstrate the robustness of our approach in handling diverse noise conditions.

### 4.3. Ablation Study on Likelihood Cost $c_l$

We conduct an ablation study on our cost function $c(\cdot, \cdot)$ (Eq. 9). Note that our cost consists of two terms: the *problem-agnostic quadratic cost $c_q$* and the *problem-adaptive likelihood cost $c_l$*. We analyze the contribution of each term by excluding each term from our cost function. The PSNR results on linear inverse problem are provided in Fig. 4b (See Table 8 in Appendix E.2 for full metric table). Our full model with both cost terms achieves the best results across most quantitative metrics. This result shows that the theoretical benefit of including the quadratic cost—namely, guaranteeing the existence and uniqueness of an OT map (Prop. 3.1)—also leads to practical performance improvement. Interestingly, the quadratic-cost-only variant remains competitive and even surpasses OTUR. This suggests that our quadratic-cost-only formulation has potential as an effective blind inverse problem solver when the corruption operator $A$ preserves the signal structure, such as in Gaussian deblurring.

## 5. Conclusion

We proposed UOTIP, an unpaired inverse problem solver model based on the Unbalanced Optimal Transport (UOT) formulation. We formulated the inverse problem as learning

the (Unbalanced) OT Map between the distributions of noisy measurements and target signals, and introduced a novel likelihood cost function. Our model achieves competitive performance on unpaired inverse problems, outperforming existing approaches. Moreover, our UOT formulation provides key advantages over the standard OT formulation, such as robustness to multi-level observation noise and class imbalance. One limitation of our work is that our method is only tested under fixed-form cost functions, without any training capacity. While our Gaussian-based likelihood cost showed some generalization to other noise types, a broader cost design could further improve performance.

## Acknowledgments

Jaewoong was supported by the National Research Foundation of Korea (NRF) grant funded by the Korea government (MSIT) [RS-2024-00349646].

## Impact Statement

This paper presents work whose goal is to advance the field of Machine Learning. There are many potential societal consequences of our work, none which we feel must be specifically highlighted here.

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

# A. Existence of the OT Map: Twist Condition

In this section, we provide the theoretical foundations for the existence and uniqueness of the Optimal Transport (OT) map within our framework. We begin by defining the **twist condition**, a standard requirement in OT theory to ensure the existence of the optimal transport map.

**Definition A.1** (Left twist condition)**.** Let $M$ and $N$ be a $n$-dimensional manifold and $N$ be a Polish space. Let $c : M \times N \to \mathbb{R}$ be a cost function and $\mu$ and $\nu$ be two probability measures on $M$ and $N$, respectively. For a given cost function $c(\mathbf{y}, \mathbf{x})$, we define the *skew left Legendre transform* as the partial map

$$\Lambda_c^l : M \times N \to T^* M, \quad \Lambda_c^l(\mathbf{y}, \mathbf{x}) = \left( \mathbf{y}, \frac{\partial c}{\partial \mathbf{y}}(\mathbf{y}, \mathbf{x}) \right)$$

whose domain of definition is

$$\mathcal{D}(\Lambda_c^l) = \left\{ (\mathbf{y}, \mathbf{x}) \in M \times N : \frac{\partial c}{\partial \mathbf{y}}(\mathbf{y}, \mathbf{x}) \text{ exists} \right\}.$$

We say that $c$ satisfies the *left twist condition* if $\Lambda_c^l$ is injective on $\mathcal{D}(\Lambda_c^l)$.

The following theorem establishes the conditions under which a deterministic optimal transport map is guaranteed to exist and be unique.

**Theorem A.2** ((Fathi & Figalli, 2010))**.** *Let $M$ be a smooth (second countable) manifold, and $N$ be a Polish space, and consider $\mu$ and $\nu$ (Borel) probability measures on $M$ and $N$ respectively. Assume that the cost $c : M \times N \to \mathbb{R}$ is lower semicontinuous and bounded from below. Moreover, assume that the following conditions hold:*

1. *the family of maps $\mathbf{y} \mapsto c(\mathbf{y}, \mathbf{x}) = c_{\mathbf{x}}(\mathbf{y})$ is locally semi-concave in $\mathbf{y}$ locally uniformly in $\mathbf{x}$,*

2. *the cost $c$ satisfies the left twist condition,*

3. *the measure $\mu$ gives zero mass to countably $(n-1)$-Lipschitz sets,*

4. *the infimum in the Kantorovitch problem $C(\mu, \nu) = \arg\min_{\gamma \in \prod} \{ \int c(\mathbf{y}, \mathbf{x}) d\gamma \}$ is finite.*

*Then there exists a borel map $T : M \to N$, which is an optimal transport map from $\mu$ to $\nu$ for the cost $c$. Moreover, the map $T$ is unique $\mu$-a.e., and any plan $\gamma_c \in \prod(\mu, \nu)$ optimal for the cost $c$ is concentrated on the graph of $T$.*

*Proof.* See (Fathi & Figalli, 2010). $\square$

## A.1. Analysis of the UOTIP Cost Function

We denote our combined cost function as $c(\mathbf{y}, \mathbf{x}; \lambda) = c_l(\mathbf{y}, \mathbf{x}) + \lambda c_q(\mathbf{y}, \mathbf{x})$ for $\lambda \geq 0$. Note that under our mild assumption in Section 2, our formulation satisfies conditions 1, 3, and 4 of Theorem A.2. However, satisfying the **left twist condition (condition 2)** is non-trivial for inverse problems.

To check the left twist condition, it is enough to show the injectivity of $\frac{\partial}{\partial \mathbf{y}} c(\mathbf{y}, \mathbf{x}; \lambda)$ with respect to $\mathbf{x}$. However, for $\lambda = 0$ (i.e., when only the likelihood-based cost term is used), the map $\mathbf{x} \mapsto \frac{\partial}{\partial \mathbf{y}} c(\mathbf{y}, \mathbf{x}; 0) = 2(\mathbf{y} - \mathcal{A}(\mathbf{x}))$ is not injective due to the ill-posedness of $\mathcal{A}$.

By incorporating the quadratic cost $c_q$, we can restore injectivity and ensure the existence of a unique solver.

**Proposition A.3.** *Assume that the corruption operator $\mathcal{A}$ is $L$-Lipschitz continuous. Then the map $\mathbf{x} \mapsto \frac{\partial}{\partial \mathbf{y}} c(\mathbf{y}, \mathbf{x}; \lambda)$ is injective for all $\lambda > L$.*

*Proof.* Note that the following equation holds:

$$\frac{\partial}{\partial \mathbf{y}} c(\mathbf{y}, \mathbf{x}; \lambda) = 2(\mathbf{y} - \mathcal{A}(\mathbf{x})) + 2\lambda(\mathbf{y} - \mathbf{x}) = (2 + 2\lambda)\mathbf{y} - 2(\lambda \mathbf{x} + \mathcal{A}(\mathbf{x})). \tag{14}$$

Thus it is enough to show that $\lambda \mathbf{x} + \mathcal{A}(\mathbf{x})$ is injective. Also, for any $\mathbf{x}_1, \mathbf{x}_2 \in \mathcal{X}$,, $\lambda \mathbf{x}_1 + \mathcal{A}(\mathbf{x}_1) = \lambda \mathbf{x}_2 + \mathcal{A}(\mathbf{x}_2)$ implies that $\lambda \|\mathbf{x}_1 - \mathbf{x}_2\| = \|\mathcal{A}(\mathbf{x}_1) - \mathcal{A}(\mathbf{x}_2)\| \leq L\|\mathbf{x}_1 - \mathbf{x}_2\|$. Thus letting $\lambda > L$, the above result implies that $\mathbf{x}_1 = \mathbf{x}_2$ and $\lambda \mathbf{x} + \mathcal{A}(\mathbf{x})$ is injective. $\square$

## B. Implementation Details

All models introduced in this paper are trained for 60,000 iterations, and we report the best results with respect to FID, even if they occur at intermediate iterations.

**Ours** In our model, unless otherwise specified, the settings follow those of UOTM (Choi et al., 2023) on CelebA-256. Our framework jointly learns the potential $v_\phi$ and the Optimal Transport Map $T_\theta$. For the potential $v_\phi$, we adopt the potential architecture from UOTM (Choi et al., 2023), while for OT Map $T_\theta$, we employ the generator architecture from OTUR (Wang et al., 2022). The learning rate for the potential is $lr_{v_\phi} = 5.0 \times 10^{-5}$, and the learning rate for the OT Map is $lr_{T_\theta} = 1.0 \times 10^{-4}$. The cost intensity hyperparameter $\tau$ is fixed to 0.001. The batch size during training is fixed at 32. The convex conjugate $\Psi_i^*$ is derived from the generator function of the KL divergence,

$$\Psi(x) = \begin{cases} x \log x - x + 1, & \text{if } x > 0 \\ \infty, & \text{if } x \le 0 \end{cases}, \tag{15}$$

which defines the associated $f$-divergence and yields the explicit form $\Psi_i^*(t) = e^t - 1$. In the case of Ours-OT, we instead set $\Psi_i^*(\cdot) = Identity(\cdot)$, while keeping all other configurations identical to those of Ours.

**Baselines** For NOT (Korotin et al., 2023), we employ the generator and discriminator of UOTM and adopt its hyperparameter settings. The batch size is set to 32. For OTUR (Wang et al., 2022) and RCOT (Tang et al., 2024), we strictly follow all configurations as proposed in their original models.

**Dataset and Evaluation** For FFHQ, we use 6,000 images from the original dataset as the clean signal, allocating 5,500 to the training set and 500 to the test set. For AFHQ, we use the original training set and the original test set as the clean signal. The measurements are generated by applying the forward operator (Eq. 1) to these clean signals. Note that under the unpaired assumption, mini-batches of measurement $Y$ and clean signal $X$ are always independently sampled in Algorithm 1. For evaluation, PSNR and SSIM are computed on the test set, whereas FID is evaluated using both the training and test sets.

**Inverse Problem** We follow the experimental settings of (Kawar et al., 2022; Song et al., 2023) for super-resolution and (Zhang et al., 2025) for the other three tasks. Formally,

- **Gaussian deblurring** (kernel size $61 \times 61$ and kernel standard deviation 3.0):

$$\mathbf{y} = \mathbf{k} * \mathbf{x} + \mathbf{n}, \ \mathbf{n} \sim \mathcal{N}(\mathbf{0}, \sigma_y^2 \mathbf{I}_m) \tag{16}$$

  where $k$ is the Gaussian kernel and $*$ denotes the convolution operator.

- **Super-resolution** ($4 \times 4$ patch downsampling):

$$\mathbf{y} = \mathbf{x} \downarrow_4 + \mathbf{n}, \ \mathbf{n} \sim \mathcal{N}(\mathbf{0}, \sigma_y^2 \mathbf{I}_m) \tag{17}$$

- **High Dynamic Range (HDR) reconstruction** (scale factor 2.0, clipping to $[-1, 1]$):

$$\mathbf{y} = \text{clip}(2\mathbf{x}, -1, 1) + \mathbf{n}, \ \mathbf{n} \sim \mathcal{N}(\mathbf{0}, \sigma_y^2 \mathbf{I}_m) \tag{18}$$

- **Nonlinear deblurring** ($\mathcal{A}$: neural operator from (Tran et al., 2021)):

$$\mathbf{y} = \mathcal{A}(\mathbf{x}) + \mathbf{n}, \ \mathbf{n} \sim \mathcal{N}(\mathbf{0}, \sigma_y^2 \mathbf{I}_m) \ \text{ for pretrained operator } \mathcal{A} \tag{19}$$

  Here, $\mathcal{A}$ is a pretrained neural operator model on the GoPro dataset (Nah et al., 2017), which learns to approximate the nonlinear blur characteristics observed in the dataset (Tran et al., 2021).

**Multi-level Observation Noise Dataset** We construct a multi-level observation noise dataset from FFHQ by first applying a corruption operator and subsequently adding noise drawn from a mixture of four levels: $\sigma_{\mathbf{y}} \in \{0.025, 0.05, 0.1, 0.2\}$. For each noise level, we randomly and independently sample degraded images in proportions of $4 : 3 : 2 : 1$. For evaluation, we generate a test set by applying all four noise levels to each of 500 FFHQ images, resulting in 2,000 degraded samples.

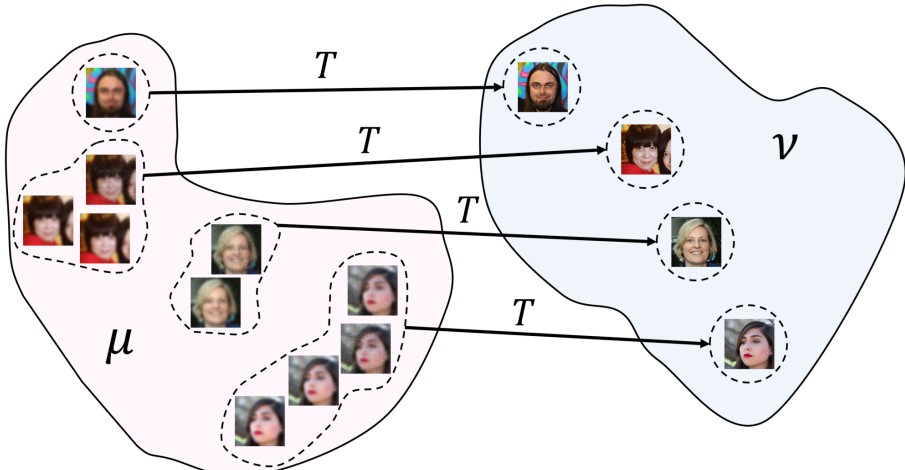

*Figure 5.* **Conceptual illustration of the multi-level observation noise problem**. A single target signal $\mathbf{x}$ may be observed under multiple Gaussian noise levels. Unlike the OT map, the UOT map can rescale each sample, enabling the alignment of multiple noisy observations with a single target signal. This flexibility makes our UOT-based model more robust to multi-level observation noise.

---

**Algorithm 1** Training algorithm of UOTIP

---

**Require:** The noisy measurement distribution $\mu$. The target image distribution $\nu$. $\Psi_i^*(t) = e^t - 1$. Generator network $T_\theta$ and the discriminator network $v_\phi$. Total iteration number $K$.

1: **for** $k = 0, 1, 2, \ldots, K$ **do**
2:     Sample a batch $Y \sim \mu$, $X \sim \nu$.
3:     $\mathcal{L}_T = \frac{1}{|Y|} \sum_{y \in Y} c(y, T_\theta(y)) - v_\phi(T_\theta(y))$
4:     Update $\theta$ by minimizing the loss $\mathcal{L}_T$.
5:     $\mathcal{L}_v = \frac{1}{|Y|} \sum_{y \in Y} \Psi_1^*(-c(y, T_\theta(y)) + v_\phi(T_\theta(y))) + \frac{1}{|X|} \sum_{x \in X} \Psi_2^*(-v_\phi(x))$
6:     Update $\phi$ by minimizing the loss $\mathcal{L}_v$.
7: **end for**

---

**Diverse Noise Types Setting**    For Laplace noise, the scale parameter is chosen such that its variance matches that of the Gaussian noise with $\sigma_{\mathbf{y}} = 0.05$. For Poisson noise, we follow the experimental setup described in (Chung et al., 2022). Formally,

- **Laplace noise** (Laplace noise scale $b = \frac{0.05}{\sqrt{2}}$):

$$\mathbf{y} = \mathcal{A}(\mathbf{x}) + \mathbf{n}, \ \mathbf{n} \sim \text{Laplace}(\mathbf{0}, b) \tag{20}$$

- **Poisson noise** (Poisson noise level $\lambda = 1.0$):

$$p(\mathbf{y}|\mathbf{x}_0) = \prod_{j=1}^{n} \frac{[A(\mathbf{x}_0)]_j^{y_j} \exp\{-[A(\mathbf{x}_0)]_j\}}{y_j!}, \tag{21}$$

where $j$ indexes the measurement bin, i.e., $j \in \{0, 1, \ldots, 255\}$. To be more specific, the Poisson noise is defined on integer pixel values $[0, 255]$. Thus, each normalized image (range $[-1, 1]$) is converted to 8-bit $[0, 255]$, Poisson noise is applied, and the result is rescaled back to $[-1, 1]$ to form the measurements.

**Algorithm**    Algorithm 1 describes the training process used in UOTIP. The generator $T_\theta$ learns to transport noisy measurements toward the target distribution by minimizing the cost term, while the discriminator $v_\phi$ is trained using the convex conjugate $\Psi_i^*(x) = e^x - 1$. By alternating these updates, the model progressively refines the transport map.

## C. Related Work

Inverse problems have been investigated from diverse approaches, ranging from traditional methods, such as hand-crafted priors and sparse coding, to deep learning techniques that learn complex image representations from large datasets.

Hand-crafted priors characterize natural images using predefined structures and simple statistical properties, such as sparsity (Candès & Wakin, 2008), low-rank representations (Fazel et al., 2008), or total variation (Candès et al., 2006). Although these methods do not require training data, they struggle to capture the complexity of natural image distributions and often fail to reliably distinguish realistic structures from unnatural artifacts. Sparse coding and dictionary learning partially mitigate this limitation by learning dictionaries from training data, but remain limited in capturing complex image structures (Qayyum et al., 2022).

With the success of deep learning, supervised approaches have achieved notable advances in many inverse problems, including image denoising (Zhang et al., 2017), super-resolution (Dong et al., 2014), and motion deblurring (Nah et al., 2017). These methods directly learn mappings from corrupted observations to clean ground-truth signals using paired datasets. However, these supervised methods rely on large collections of paired noisy/clean images, which are expensive or infeasible to obtain in many domains. Moreover, they often generalize poorly to out-of-distribution measurements (Recht et al., 2019).

To address these challenges, unsupervised and self-supervised approaches have been proposed that eliminate the need for paired data by leveraging measurement consistency—enforcing that the reconstructed image $\hat{x}$, when passed through the forward operator $A$, reproduces the observed measurements $y$ (Ulyanov et al., 2018). More recently, generative models have emerged as powerful priors, such as GANs (Bora et al., 2017), VAEs (Goh et al., 2019), and diffusion models (Song et al., 2021; Chung et al., 2022; Zhang et al., 2025). These models learn the distribution of natural images, enabling reconstruction by combining learned priors with measurement consistency. While these approaches achieve high perceptual quality, they often require expensive pretraining and remain sensitive to mismatches in noise assumptions.

Among generative models, Neural Optimal Transport models (Fan et al., 2023; Rout et al., 2022; Choi et al., 2024a; 2023; 2025a; Choi & Choi, 2024) are particularly relevant to our work. The prior works (Korotin et al., 2023; Tang et al., 2024) adopted OT-map-based approaches for inverse problems. Our work is the first attempt to generalize the OT framework to the Unbalanced Optimal Transport (UOT) framework and introduce the likelihood cost function. While learned regularizer approaches (Lunz et al., 2018; Goujon et al., 2023) share a superficial similarity with OT-map–based approaches (Korotin et al., 2023; Tang et al., 2024) on the training loss forms, their underlying formulations differ fundamentally. Lunz et al. (2018) incorporate a WGAN regularizer into the measurement loss, requiring a 1-Lipschitz discriminator and leading to a min–max optimization over the generator and discriminator. Goujon et al. (2023) further adopt a proximal-gradient reconstruction scheme utilizing the learned regularizer. In contrast, the OT-map-based approaches (Korotin et al., 2023; Tang et al., 2024) derive the training objective based on the semi-dual OT formulation. This results in a similar loss function **without Lipschitz constraints on the potential function, and the training becomes a max–min problem** that replaces the measurement loss with a quadratic cost function. This difference leads to fundamentally different theoretical properties, including the existence and uniqueness of solutions (Choi et al., 2025b) and the interpretation of the learned potential (Fan et al., 2023), and different practical performance (Choi et al., 2024b). Importantly, the OT potential is defined between the **measurement distribution and the signal distribution**, whereas the GAN discriminator operates between the **true signal distribution and the generated distribution**, making the two notions conceptually and mathematically different.

In this paper, we introduce an inverse problem solver based on Unbalanced Optimal Transport (UOT). Unlike many existing methods, our method requires neither paired training data nor pretraining on large-scale natural image datasets. By incorporating a likelihood-based cost, our approach directly aligns reconstruction with the MAP estimate. Theoretically, the UOT framework relaxes the strict marginal-matching constraint, providing robustness to outliers, adaptability to multi-level observation noise and class imbalance, and generalization across diverse noise conditions. These properties make our method particularly suited for unpaired inverse problems, where only independent sets of noisy and clean samples are available.

## D. Additional Qualitative Examples

### D.1. Comparison of inverse problem solvers on AFHQ

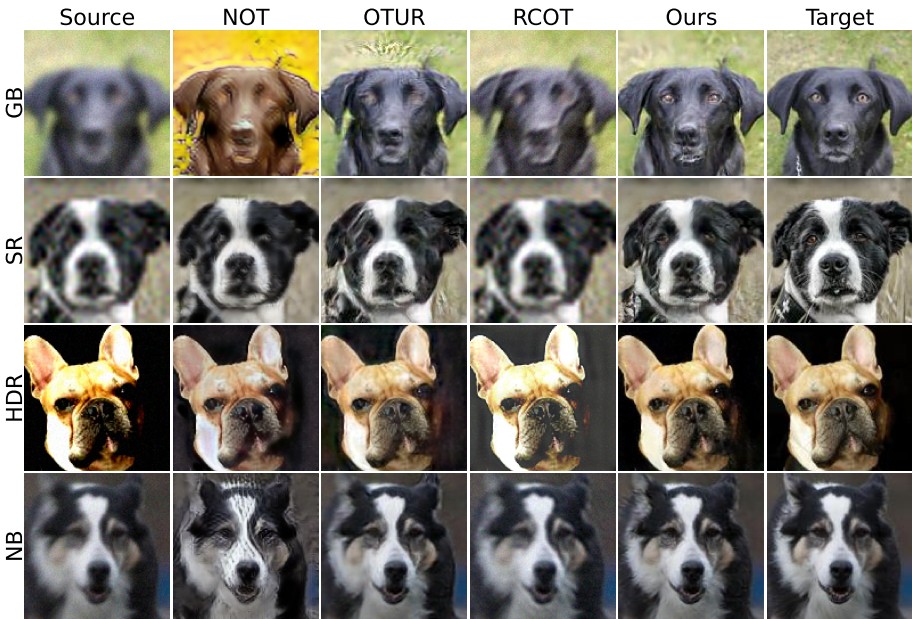

*Figure 6.* **Comparison of inverse problem solvers on AFHQ** for four tasks: Gaussian deblurring (GB), Super-resolution (SR), High dynamic range reconstruction (HDR), and Nonlinear deblurring (NB). Our model reconstructs images of superior fidelity with well-preserved structural details.

## D.2. Results for Various Inverse Problems

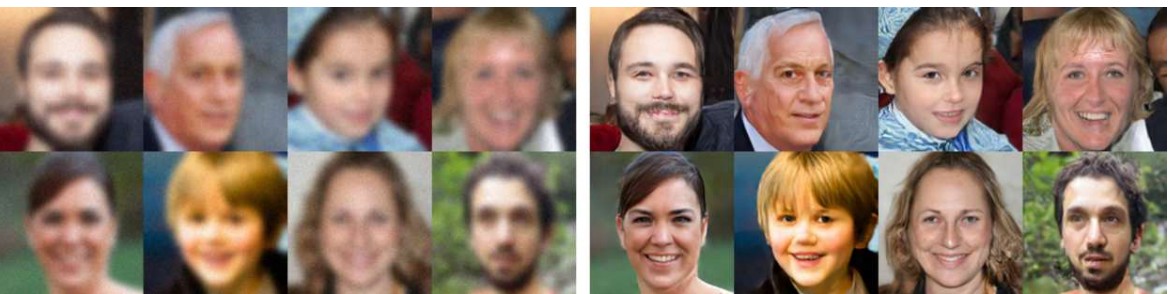

*Figure 7.* Additional qualitative results of our model for the Gaussian deblurring task on FFHQ (degraded (Left) → clean (Right)).

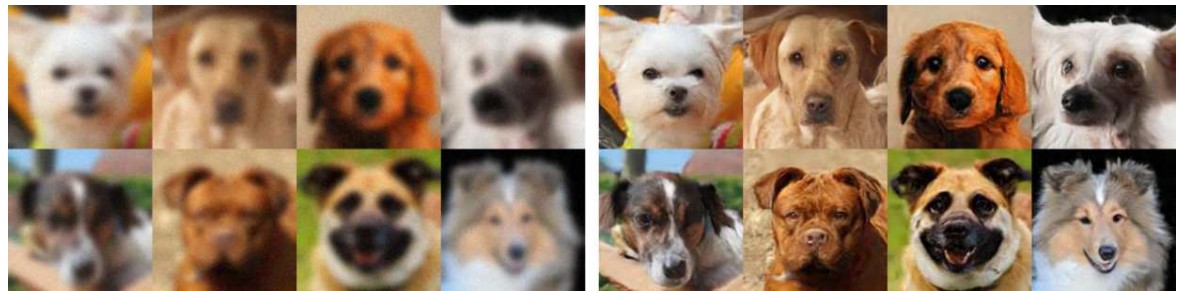

*Figure 8.* Additional qualitative results of our model for the Gaussian deblurring task on AFHQ (degraded (Left) → clean (Right)).

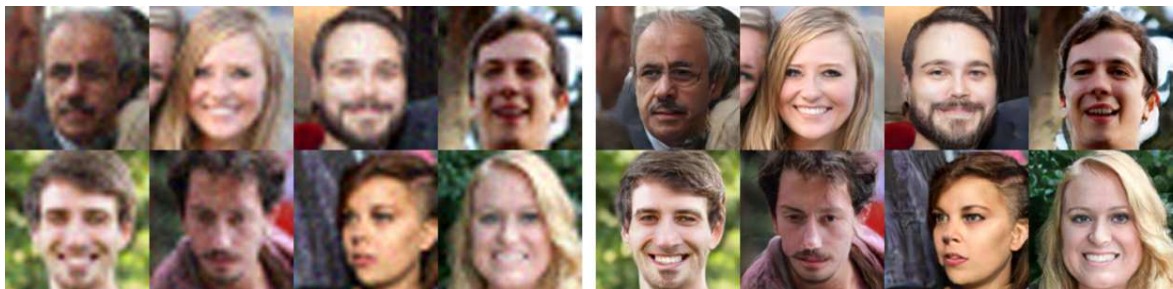

*Figure 9.* Additional qualitative results of our model for the super-resolution task on FFHQ (degraded (Left) → clean (Right)).

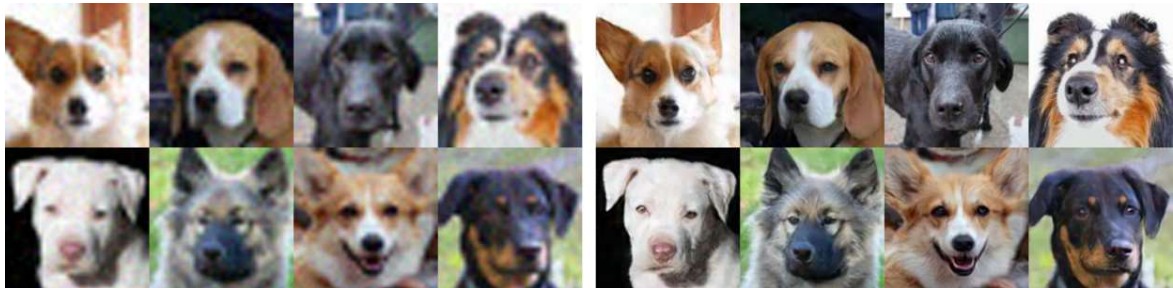

*Figure 10.* Additional qualitative results of our model for the super-resolution task on AFHQ (degraded (Left) → clean (Right)).

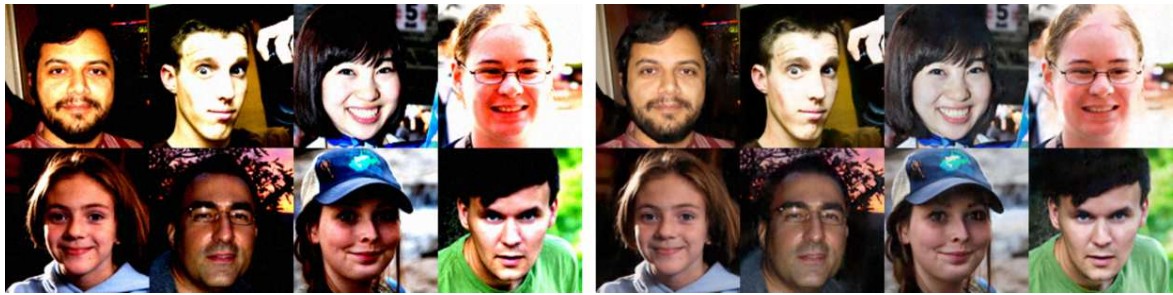

*Figure 11.* Additional qualitative results of our model for the HDR reconstruction task on FFHQ (degraded (Left) → clean (Right)).

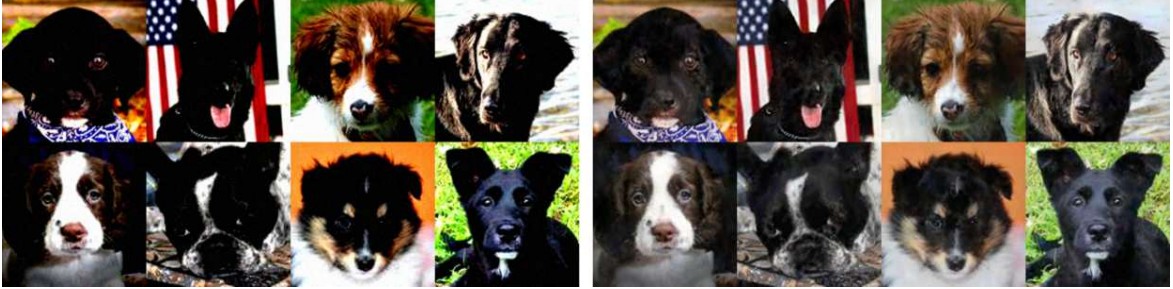

*Figure 12.* Additional qualitative results of our model for the HDR reconstruction task on AFHQ (degraded (Left) → clean (Right)).

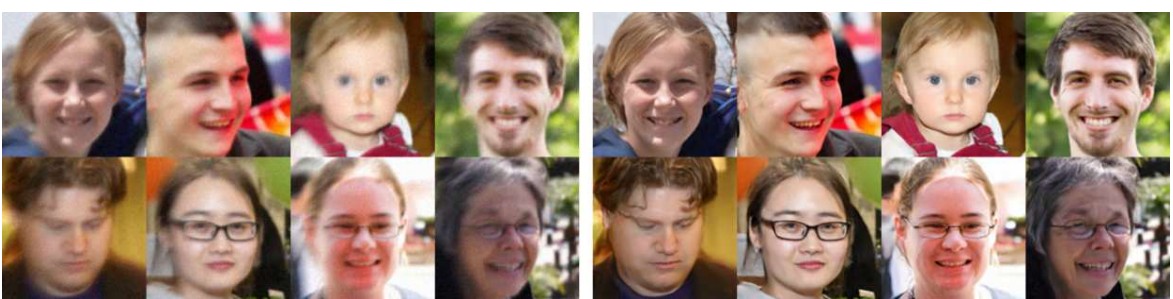

*Figure 13.* Additional qualitative results of our model for the nonlinear deblurring task on FFHQ (degraded (Left) → clean (Right)).

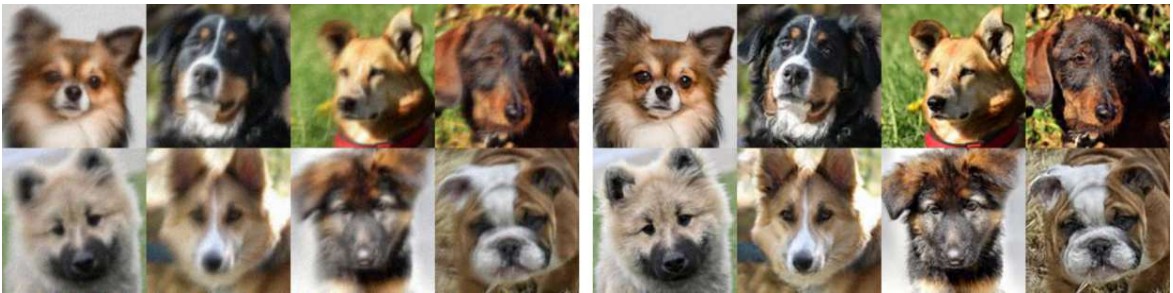

*Figure 14.* Additional qualitative results of our model for the nonlinear deblurring task on AFHQ (degraded (Left) → clean (Right)).

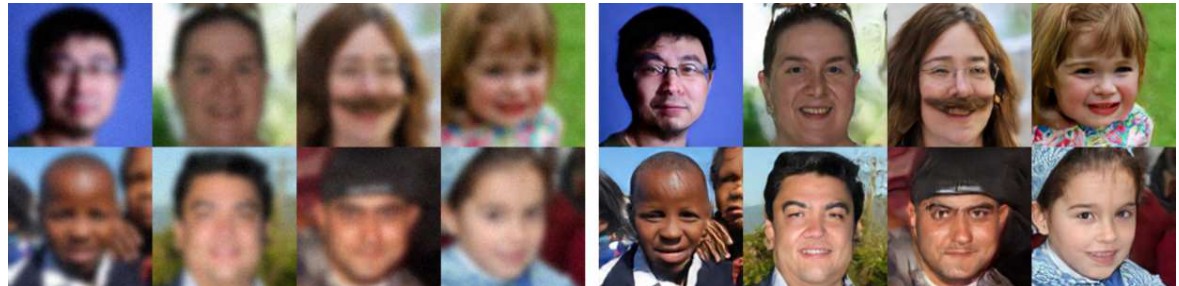

*Figure 15.* Qualitative results of our quadratic-cost-only variant for the Gaussian deblurring task on FFHQ (degraded (Left) → clean (Right)).

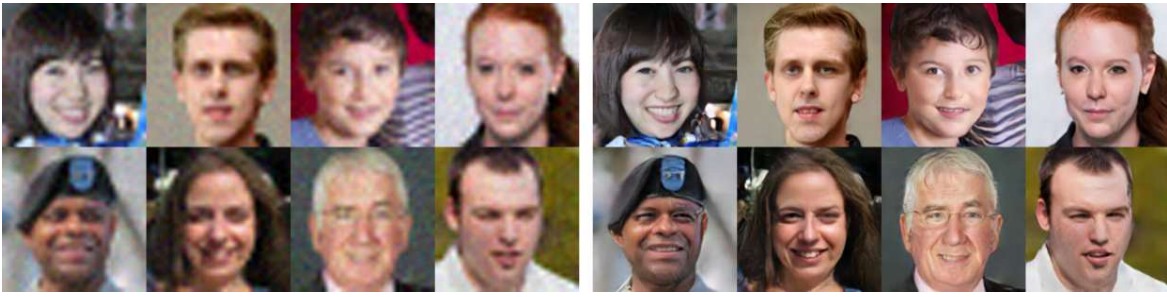

*Figure 16.* Qualitative results of our quadratic-cost-only variant for the super resolution task on FFHQ (degraded (Left) → clean (Right)).

### D.3. Results for Multi-level observation noise

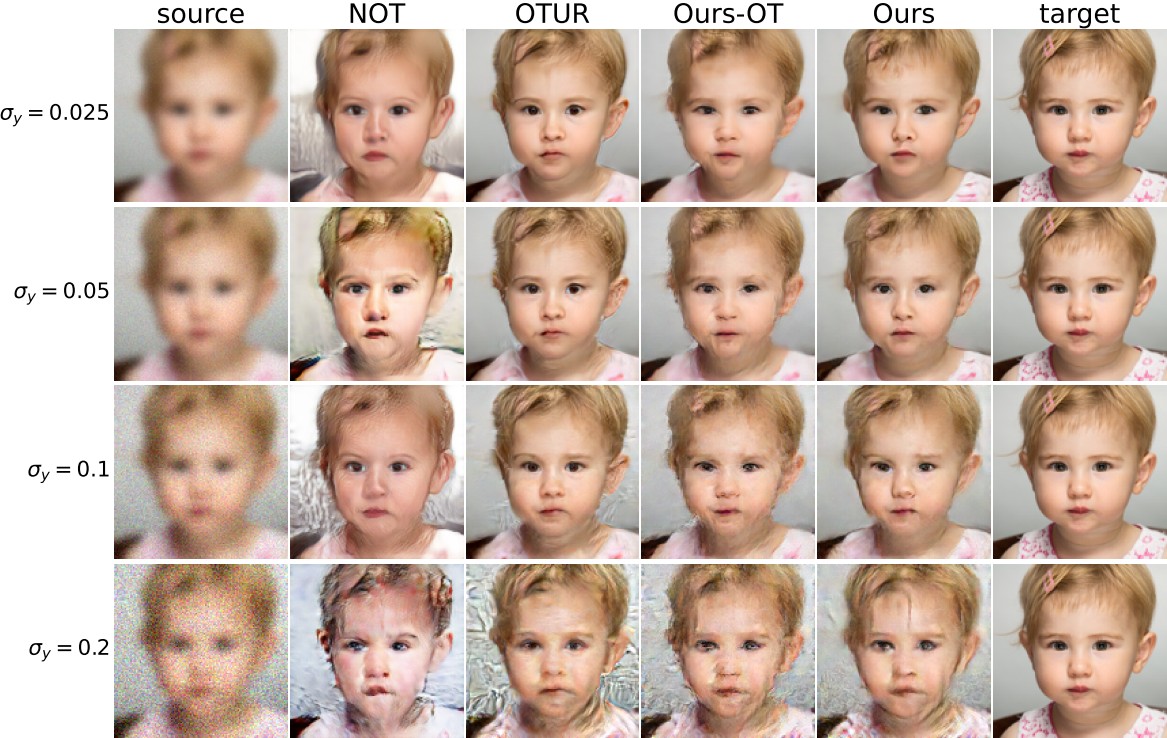

*Figure 17.* **Comparison of inverse problem solvers under multi-level observation noise** on FFHQ for the Gaussian deblurring.

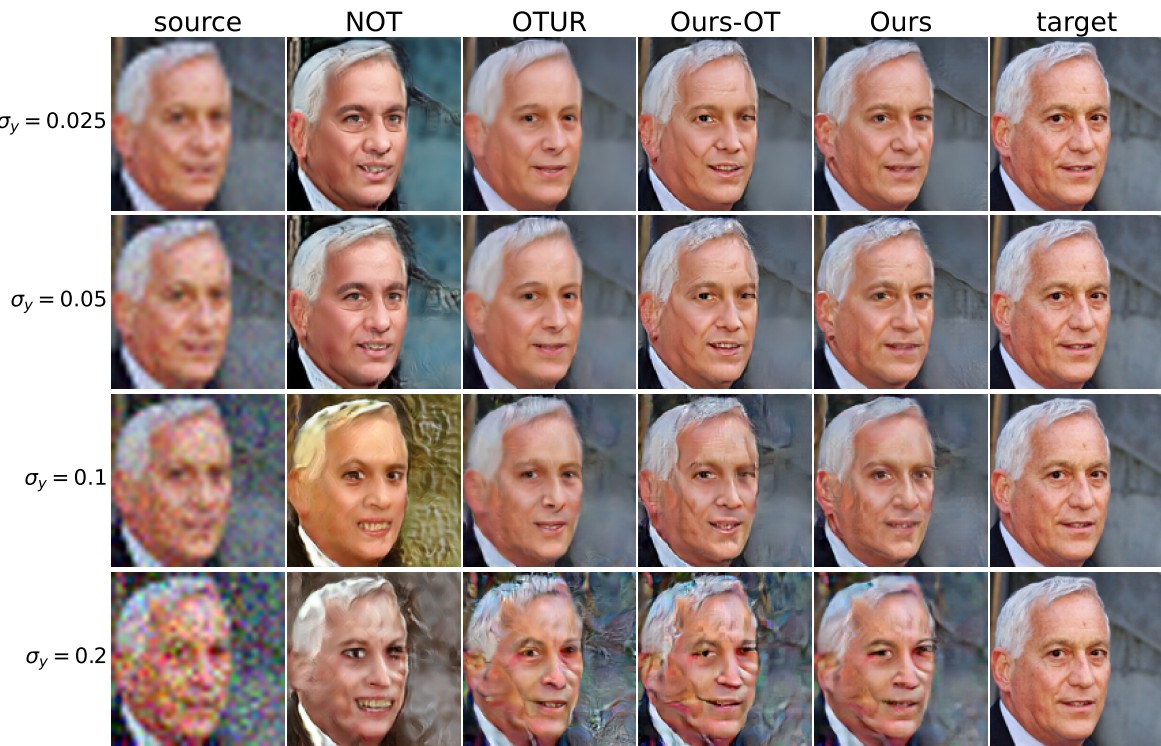

*Figure 18.* **Comparison of inverse problem solvers under multi-level observation noise** on FFHQ for the super-resolution.

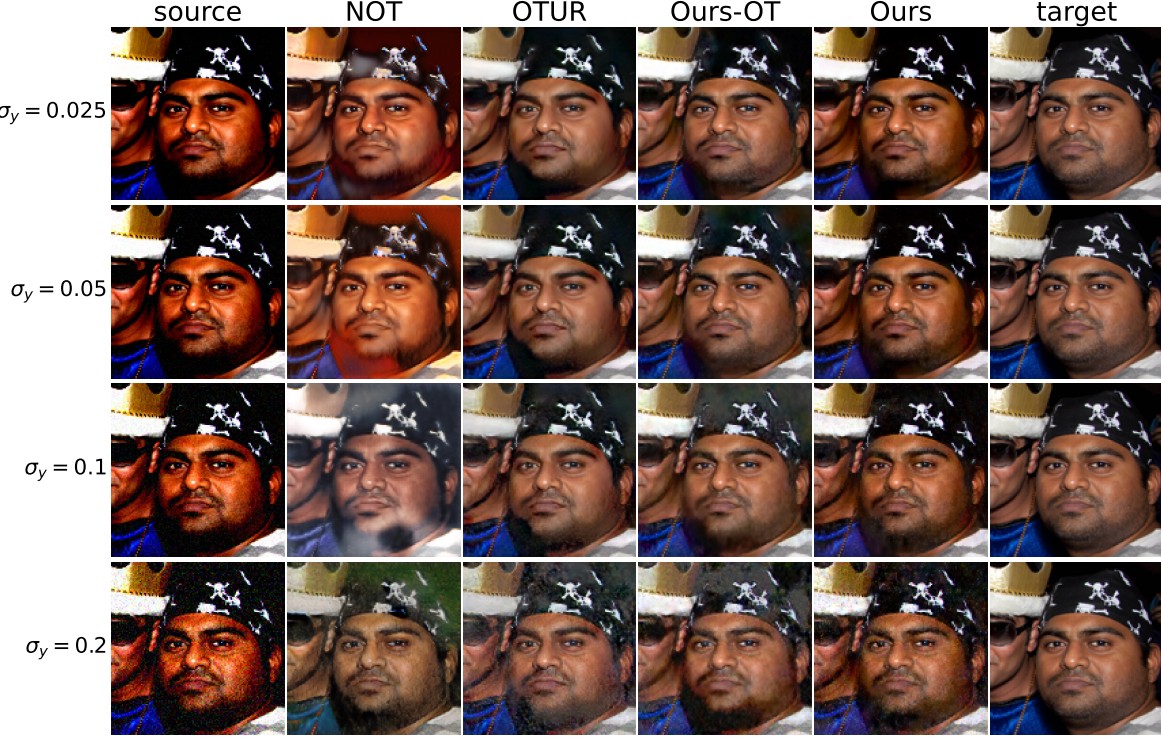

*Figure 19.* **Comparison of inverse problem solvers under multi-level observation noise** on FFHQ for the HDR reconstruction.

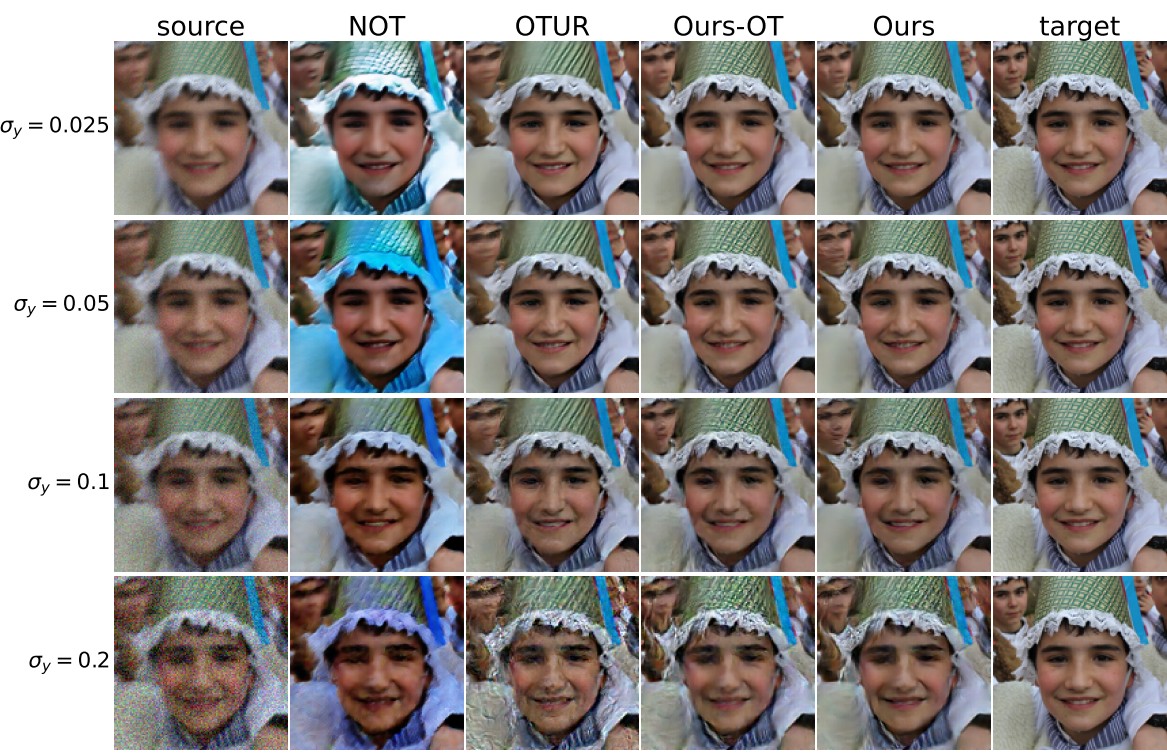

*Figure 20.* **Comparison of inverse problem solvers under multi-level observation noise** on FFHQ for the nonlinear deblurring.

# E. Additional Quantitative Results

### E.1. Discussion on the Cost Intensity Parameter $\tau$

**Theoretical condition**   In our implementation, we do not explicitly enforce the condition $\lambda > L$. Instead, the role of $\lambda$ in the theoretical analysis is reflected through the hyperparameter $\tau$ used in our cost function (Eq. 9), which scales the intensity of the cost function. Although this strategy performs reliably in practice, investigating a more principled strategy for adapting $\tau$ remains an interesting direction for future work.

**Results for cost intensity parameter $\tau$ ablation**   We perform an ablation study on the cost intensity hyperparameter $\tau$ for two linear inverse problems. On FFHQ dataset, we tested values $\tau \in 0.001 \times \{0.25, 0.5, 1, 2, 4\}$ (Fig. 21). Across both Gaussian deblurring and super-resolution, our model remains robust to the choice of $\tau$, except when $\tau$ is excessively large. Specifically, metrics that directly compare with ground-truth signals (PSNR and SSIM) remain stable. On the other hand, the metric that measures marginal distribution-level fidelity (FID) degrades notably as $\tau$ increases. See Table 3 for the full result table. To further examine sensitivity across datasets, we additionally conduct experiments on AFHQ for Gaussian deblurring. The results show that our method remains stable across datasets, with PSNR varying only marginally as $\tau$ changes.

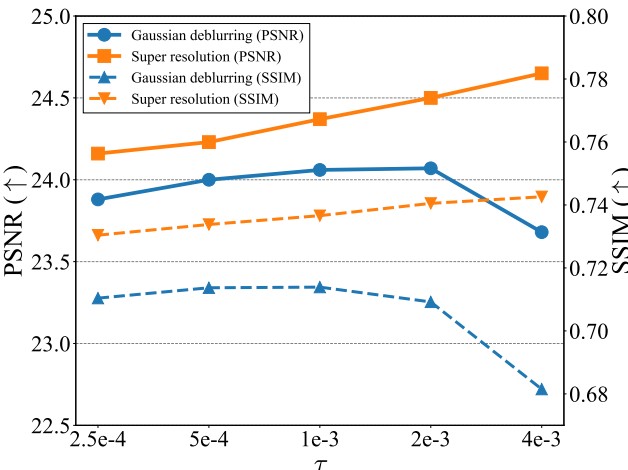

*Figure 21.* **Ablation study on the cost intensity parameter** $\tau$. For clarity, only PSNR ($\uparrow$) and SSIM ($\uparrow$) are visualized.

*Table 3.* Ablation study on the cost intensity parameter $\tau$ with the FFHQ dataset. $\tau_0 = 1e - 3$.

| $\tau$ | Gaussian Deblurring | | | Super Resolution | | |
|---|---|---|---|---|---|---|
| | PSNR ($\uparrow$) | SSIM ($\uparrow$) | FID ($\downarrow$) | PSNR ($\uparrow$) | SSIM ($\uparrow$) | FID ($\downarrow$) |
| $0.25 \times \tau_0$ | 23.88 | 0.7104 | **18.580** | 24.16 | 0.7304 | **18.708** |
| $0.5 \times \tau_0$ | 24.00 | 0.7137 | 19.093 | 24.23 | 0.7338 | 18.824 |
| $\tau_0$ | 24.06 | **0.7139** | 21.210 | 24.37 | 0.7366 | 20.234 |
| $2 \times \tau_0$ | **24.07** | 0.7092 | 30.739 | 24.50 | 0.7405 | 21.277 |
| $4 \times \tau_0$ | 23.68 | 0.6815 | 74.994 | **24.65** | **0.7426** | 30.281 |

*Table 4.* Sensitivity analysis on the cost intensity parameter $\tau$ across datasets for Gaussian deblurring.

| Dataset | $\tau_0 = 0.001$ | PSNR ($\uparrow$) | FID ($\downarrow$) |
|---|---|---|---|
| | $\times 0.5$ | 24.18 | 10.652 |
| AFHQ | $\times 1$ | 24.22 | 12.566 |
| | $\times 2$ | 24.26 | 19.328 |
| | $\times 0.5$ | 24.00 | 19.093 |
| FFHQ | $\times 1$ | 24.06 | 21.210 |
| | $\times 2$ | 24.07 | 30.739 |

## E.2. Quantitative Result Tables

*Table 5.* **Quantitative results under multi-level observation noise** for four inverse problems on FFHQ. Our model exhibits superior robustness across noise levels.

*(a)* Nonlinear inverse problems: High Dynamic Range and Nonlinear Deblurring.

| Method | High Dynamic Range | | | Nonlinear Deblurring | | |
|---|---|---|---|---|---|---|
| | PSNR ($\uparrow$) | SSIM ($\uparrow$) | FID ($\downarrow$) | PSNR ($\uparrow$) | SSIM ($\uparrow$) | FID ($\downarrow$) |
| NOT (Korotin et al., 2023) | 21.01 | 0.7728 | 62.641 | 21.50 | 0.7201 | 76.795 |
| OTUR (Wang et al., 2022) | 24.29 | 0.8098 | 52.149 | 25.23 | 0.7763 | 61.244 |
| OTIP (Ours-OT) | **25.92** | **0.8488** | **49.750** | 26.90 | 0.8307 | 51.805 |
| UOTIP (Ours) | 25.44 | 0.8330 | 51.300 | **27.25** | **0.8427** | **43.450** |

*Table 6.* **Class imbalance results under imbalance ratio** $k$ between AFHQ-cat and AFHQ-dog for the Gaussian deblurring.

| Ratio | Model | PSNR ($\uparrow$) | SSIM ($\uparrow$) | FID ($\downarrow$) |
|---|---|---|---|---|
| 1 | NOT | 20.28 | 0.5567 | 50.166 |
| | OTUR | 23.13 | 0.6477 | 28.626 |
| | OTIP (Ours-OT) | 23.15 | 0.6373 | 28.182 |
| | UOTIP (Ours) | **23.42** | **0.6523** | **21.558** |
| 2 | NOT | 19.10 | 0.5073 | 60.739 |
| | OTUR | 22.92 | 0.6435 | 32.429 |
| | OTIP (Ours-OT) | 22.64 | 0.6190 | 34.047 |
| | UOTIP (Ours) | **23.29** | **0.6438** | **25.689** |
| 3 | NOT | 18.35 | 0.4977 | 66.060 |
| | OTUR | 22.19 | 0.6182 | 36.812 |
| | OTIP (Ours-OT) | 22.50 | 0.6128 | 66.842 |
| | UOTIP (Ours) | **23.04** | **0.6315** | **31.630** |
| 4 | NOT | 18.11 | 0.4674 | 58.237 |
| | OTUR | 20.92 | 0.5671 | 40.651 |
| | OTIP (Ours-OT) | 22.42 | 0.6045 | 78.869 |
| | UOTIP (Ours) | **22.67** | **0.6155** | **39.894** |

*Table 7.* **Quantitative results under diverse noise types** in linear inverse problems on the FFHQ dataset.

| Ratio | Method | Gaussian Deblurring | | | Super Resolution $4\times$ | | |
|---|---|---|---|---|---|---|---|
| | | PSNR ($\uparrow$) | SSIM ($\uparrow$) | FID ($\downarrow$) | PSNR ($\uparrow$) | SSIM ($\uparrow$) | FID ($\downarrow$) |
| Gaussian noise | NOT | 20.11 | 0.6035 | 52.901 | 20.13 | 0.6257 | 50.066 |
| | OTUR | 23.82 | 0.7106 | 24.337 | 24.09 | 0.7243 | 22.751 |
| | UOTIP (Ours) | **24.06** | **0.7139** | **21.210** | **24.35** | **0.7371** | **19.475** |
| Laplace noise | NOT | 19.50 | 0.5810 | 57.921 | 20.41 | 0.6277 | 50.228 |
| | OTUR | 23.63 | 0.7074 | 23.825 | 23.72 | 0.7252 | 23.931 |
| | UOTIP (Ours) | 23.98 | 0.7153 | **21.846** | **24.04** | **0.7350** | **19.629** |
| | UOTIP (Ours-Laplace's likelihood) | **24.12** | **0.7204** | 25.342 | 23.99 | 0.7317 | 27.053 |
| Poisson noise | NOT | 19.15 | 0.5584 | 55.368 | 19.28 | 0.5820 | 57.606 |
| | OTUR | 23.14 | 0.6919 | 25.840 | 23.26 | 0.7020 | 26.222 |
| | UOTIP (Ours) | **23.47** | **0.6938** | **23.669** | **23.59** | **0.7163** | **20.396** |
| | UOTIP (Ours-Poisson likelihood) | 18.06 | 0.4295 | 290.053 | 22.56 | 0.6075 | 175.036 |

We conducted experiments using the likelihood cost functions for Laplace and Poisson noise in Table 7.

- For Laplace noise, we used the L1 likelihood:

$$c_{l,\text{laplace}}(\mathbf{y}, \mathbf{x}) = -\|\mathbf{y} - A\mathbf{x}\|_1 \tag{22}$$

- For Poisson noise, we used the standard scaled quadratic approximation (as in Chung et al. (2022)):

$$c_{l,\text{poisson}}(\mathbf{y}, \mathbf{x}) = -\|\mathbf{y} - A(\mathbf{x})\|_\Lambda^2, \qquad [\Lambda]_{ii} \triangleq \frac{1}{2y_i}, \qquad \|\mathbf{a}\|_\Lambda^2 \triangleq \mathbf{a}^T \Lambda \mathbf{a}. \tag{23}$$

For the Poisson-likelihood experiments reported in Table 7, we adopt the standard scaled quadratic approximation, following Chung et al. (2022). Under this approximation, the Poisson-likelihood variant performs worse than the Gaussian-likelihood variant. This behavior is consistent with the numerical instability often observed in Poisson likelihood objectives, as also discussed in Appendix C.4 of Chung et al. (2022). By contrast, under Laplace noise, the Laplace-likelihood variant achieves competitive performance. These observations indicate that our framework is not inherently tied to the Gaussian likelihood.

*Table 8.* Ablation study on the cost function $c(\mathbf{y}, \mathbf{x})$, investigated on the FFHQ dataset

| | cost term | | Gaussian Deblurring | | | Super Resolution $4\times$ | | |
|---|---|---|---|---|---|---|---|---|
| | Quad term | IP term | PSNR ($\uparrow$) | SSIM ($\uparrow$) | FID ($\downarrow$) | PSNR ($\uparrow$) | SSIM ($\uparrow$) | FID ($\downarrow$) |
| | ✓ | | 24.01 | 0.7130 | 21.516 | 24.29 | 0.7332 | 20.131 |
| Ours | | ✓ | 23.98 | **0.7140** | 25.608 | 24.22 | 0.7354 | **19.085** |
| | ✓ | ✓ | **24.06** | 0.7139 | **21.210** | **24.35** | **0.7371** | 19.475 |
| OTUR (Wang et al., 2022) | | | 23.82 | 0.7106 | 24.337 | 23.91 | 0.6777 | 30.773 |

# F. Additional Empirical Comparisons

To further strengthen the empirical evaluation, we conducted additional experiments by comparing our method with DPS (Chung et al., 2022) and the more recent OT-based method KIDOT (Zheng et al., 2026).

DPS is a relevant diffusion-based baseline that can be compared under our setting. By contrast, many other diffusion-based methods, such as DDRM (Kawar et al., 2022) and DAPS (Zhang et al., 2025), require additional knowledge of the measurement noise, in particular the noise intensity , which makes a fair direct comparison with our setting inappropriate. KIDOT, on the other hand, is designed for medical image reconstruction tasks such as MRI and CT, and relies on the adjoint of the operator, which restricts it to linear inverse problems. Accordingly, we additionally included comparisons on linear inverse problems.

As shown in Table 9, our method outperforms both DPS and KIDOT across all metrics. In particular, the large gap between DPS's test-set FID and full-dataset FID suggests memorization. These comparisons with both a non-OT baseline and a more recent OT-based method further validate the empirical effectiveness of our approach.

*Table 9.* Additional empirical comparisons on FFHQ: two linear (Gaussian deblurring, Super-resolution) and two nonlinear (HDR Reconstruction, Nonlinear Deblurring). The **boldface** values indicate the best performance, except for FID-Total, which is reported only for reference.

| Task | Method | PSNR ($\uparrow$) | SSIM ($\uparrow$) | FID-Test ($\downarrow$) | FID-Total ($\downarrow$) |
|---|---|---|---|---|---|
| Super-resolution 4× | DPS (Chung et al., 2022) | 17.00 | 0.4250 | 78.440 | 4.573 |
| | KIDOT (Zheng et al., 2026) | 22.17 | 0.5528 | 200.633 | 155.541 |
| | UOTIP (Ours) | **24.35** | **0.7371** | **52.273** | 19.475 |
| Gaussian Deblurring | DPS (Chung et al., 2022) | 18.91 | 0.4769 | 73.788 | 4.341 |
| | KIDOT (Zheng et al., 2026) | 21.98 | 0.5361 | 239.004 | 190.314 |
| | UOTIP (Ours) | **24.06** | **0.7139** | **55.869** | 21.210 |
| HDR Reconstruction | DPS (Chung et al., 2022) | 19.38 | 0.5440 | 79.358 | 4.886 |
| | UOTIP (Ours) | **26.02** | **0.8642** | **45.823** | 20.840 |
| Nonlinear Deblurring | DPS (Chung et al., 2022) | 19.01 | 0.4938 | 73.414 | 4.387 |
| | UOTIP (Ours) | **28.52** | **0.8841** | **36.974** | 11.370 |

