# OpenReview forum: "UOTIP: Unbalanced Optimal Transport Map for Unpaired Inverse Problems"
_ICML.cc/2026/Conference — ICML 2026 regular_

### Official Review · Reviewer_yhE9 · 2026-03-06

**Soundness:** 3
**Presentation:** 3
**Significance:** 2
**Originality:** 3
**Overall Recommendation:** 4
**Confidence:** 4

**Summary:**

The paper proposes Unbalanced Optimal Transport Map for Inverse Problems (UOTIP), a novel approach for solving unpaired image inverse problems. The method formulates the reconstruction of clean target signals from noisy measurements as learning an Unbalanced Optimal Transport (UOT) map. To achieve this, the authors introduce a combined cost function comprising a likelihood-based cost (for data fidelity) and a quadratic cost term. The authors evaluate their model on linear (Gaussian deblurring, super-resolution) and nonlinear (HDR reconstruction, nonlinear deblurring) tasks.

**Compliance With Llm Reviewing Policy:**

Affirmed.

**Final Justification:**

Though the performance is improved, the motivation for the additional $c_q$ and the problems to which it can be applied are not convincing.

**Key Questions For Authors:**

1. In Line 141, the authors introduce the term  $-v^{\star c}(y)$. Could the authors clarify the meaning of the superscript “c”?
2. The authors employ Equation (9) as the loss function. Does this loss function still admit a Maximum A Posteriori (MAP) interpretation? If so, please clarify the underlying assumptions.
3. I am confused about the definition of the cost function between Equation (9) and Proposition 3.1. In Equation (9), the cost function involves the parameter $\tau$, whereas in Proposition 3.1 it involves $\lambda$, together with the requirement $\lambda > L$. How is this requirement guaranteed in practice? In particular, the numerical experiments appear to fix $\tau = 0.001$. Could the authors discuss the sensitivity of the proposed method to the choice of $\tau$ across different datasets?
4. In the numerical results reported in Table 4, OTIP without the “Unbalanced” component outperforms UOTIP on the High Dynamic Range dataset. Could the authors provide an explanation for this behavior?
5. In Line 641, the authors state that the formulation satisfies Conditions 1, 3, and 4 of Theorem A.2. Could the authors briefly explain why these conditions are satisfied?

**Limitations:**

Yes

**Strengths And Weaknesses:**

**Strengths**
- **Empirical Performance:** The method demonstrates strong quantitative results (PSNR, SSIM, FID) across multiple standard image restoration benchmarks compared to existing unpaired baselines like NOT and OTUR.
- **Benefits of Unbalanced OT:** The shift from standard Optimal Transport to Unbalanced Optimal Transport is well-motivated. By relaxing strict marginal constraints, the model naturally handles class imbalances between unpaired datasets and shows robustness to multi-level observation noise.
- **Theoretical Foundations for Identical Spaces:** For problems where the input and output dimensions match, the authors provide a solid mathematical proof (Proposition 3.1) showing that the addition of the quadratic cost term ensures the "twist condition" is satisfied, guaranteeing the existence and uniqueness of the OT map for ill-posed problems

**Weaknesses**
- **Overclaimed Title and Lack of Applicability to General Inverse Problems** The title and abstract claim to introduce a solver for "Unpaired Inverse Problems". However, this claim is fundamentally overblown because the method cannot be applied to general inverse problems. The core issue lies in the required quadratic cost function, defined as $\|y-x\|_2^2$​. This formulation strictly demands that the noisy measurement $y$ and the target signal $x$ reside in the exact same dimensional space and be close.  In general inverse problems (e.g., medical imaging like MRI/CT, seismic imaging, or compressive sensing), the measurement space and the signal space are vastly different. The current UOTIP framework implicitly assumes that the input and output are spatially "close" or identical in shape, effectively limiting the model to image-to-image translation tasks like denoising or deblurring, rather than acting as a general inverse problem solver.
- **Catastrophic Loss of Theoretical Guarantees in Super-Resolution** The strict dimensional requirement of the quadratic cost forces the authors into a problematic workaround for their super-resolution benchmark. Because $y$ and $x$ have different dimensions, they apply a bicubic interpolation operator $Q$ to the lower-resolution image $y$, modifying the cost to $\|Q(y)−x\|_2^2$​. As the authors admit in a footnote, this modification completely invalidates the twist condition established in Proposition 3.1. Consequently, for any task where input and output dimensions differ, the theoretical guarantee for the existence of the OT map is entirely lost. This is a severe flaw, as the twist condition is presented as one of the paper's main theoretical contributions.
- **Failure of the Likelihood Cost Adaptability** The authors claim that their likelihood cost can be flexibly tailored to specific inverse problems. However, empirical results contradict this. Table 6 reveals a catastrophic failure when the model attempts to use the mathematically correct Poisson likelihood cost for Poisson noise (e.g., in Gaussian Deblurring, PSNR drops to 18.06 and FID explodes to 290.053). The model only achieves reasonable results on Poisson noise when it incorrectly applies the Gaussian likelihood cost. This indicates the framework is highly rigid and fundamentally tied to Gaussian assumptions, contradicting the claim of broad noise-type generalization.
- **References**: “Featured Certification.” is included in Fan et al., 2023; Choi et al., 2025c and Choi et al., 2025b are the same. Missing conference title in Tang et al., 2024. Missing publisher information for Santambrogio, 2015.

---

> ### Author Rebuttal · Authors · 2026-03-31
>
> We sincerely thank the reviewer for the careful reading and valuable feedback. Due to the rebuttal length constraint, we focused on the major comments. Please let us know if you have any additional concerns.
>
> > [W1] The title “Unpaired Inverse Problems” appears overclaimed, as the quadratic cost limits the method’s applicability to general inverse problems.
>
> **A.**  We would first like to clarify that **Unpaired Inverse Problems** refers to the absence of paired input–output data in the problem setting, and **is not intended to imply unrestricted applicability to all possible inverse problems**. In other words, “unpaired” in our paper describes the problem setting, rather than the generality of the forward operator or measurement space.
>
> Regarding the quadratic cost, it is introduced to ensure the theoretical existence and uniqueness of the solution, rather than being a fundamental requirement of the method. More broadly, if an alternative cost satisfying the twist condition can be designed, the framework can in principle be extended to other inverse problem settings. Furthermore, as shown in Table 7 and Figure 4(b), removing the quadratic term still yields competitive performance, suggesting it serves primarily as a theoretically motivated component.
>
> ---
> > [W2] Loss of the twist-condition guarantee in super-resolution due to the modified quadratic cost used to match the dimensions of $x$ and $y$.
>
> **A.** We agree with the reviewer’s concern. In the unequal-dimensional case, the point raised by the reviewer is indeed valid unless the quadratic term $||T(y)-x||_2^2$ satisfies the twist condition. However, we would like to emphasize that one of our main contributions is to approach inverse problems through optimal transport with a likelihood-based cost. In practice, Table 7 and Fig. 4(b) shows that our method still achieves competitive performance even without the quadratic term. Extending the discussion to the unequal-dimensional setting would also be a very interesting direction for future research.
>
> ---
> > [W3] Failure of likelihood-cost adaptability, as the method performs poorly with the Poisson likelihood cost and only works well when using the Gaussian likelihood cost instead.
>
> **A.**  We would like to clarify that our main purpose is to demonstrate that our framework, developed under a Gaussian noise assumption, can still generalize effectively to other noise models, rather than to modify it separately for each likelihood.
>
> For the Poisson-likelihood results in Table 6, we used the standard scaled quadratic approximation (as in DPS). In this case, using the Poisson likelihood yields worse performance than using the Gaussian likelihood. This aligns with known numerical instabilities in Poisson likelihood objectives (Appendix C.4 of DPS). By contrast, for Laplace noise, using the Laplace likelihood still produced competitive results. This provides further evidence that our framework is not strongly tied to the Gaussian assumption.
>
> ---
> > [Q3] Clarification on the discrepancy between the cost definitions in Equation (9) and Proposition 3.1, how the condition $\lambda > L$ is guaranteed in practice, and the sensitivity of the proposed method to the choice of $\tau$ across datasets.
>
> **A.**  In our implementation, we do not explicitly enforce the condition $\lambda>L$. Instead, the role of $\lambda$ in the theoretical analysis is reflected through the hyperparameter $\tau$ used in our cost function (Eq.9), which scales the intensity of the cost function. Although this method performs reliably in practice, we agree that investigating a more principled strategy for adapting $\lambda$ remains an interesting prospect for future work. For the sensitivity of our model to $\tau$ across different datasets, we additionally conducted experiments on AFHQ for the Gaussian deblurring (GD).
>
> |GD|$\tau_0 = 0.001$| PSNR ↑|FID ↓|
> |:--|--|--|--|
> |AFHQ|$\times 0.5$| 24.18|10.652|
> ||$\times 1$| 24.22|12.566|
> ||$\times 2$|24.26|19.328|
> |FFHQ|$\times 0.5$| 24.00|19.093|
> ||$\times 1$| 24.06|21.210|
> ||$\times 2$|24.07|30.739|
> ---
> > [Q5] Clarification on the satisfaction of Conditions 1, 3, and 4 in Theorem A.2
>
> **A.**  For the map $c_x(y)=||y- A(x)||_2^2$, we have
> $$c_x(\tilde{y})-c_x(y)=\langle 2y-2A(x),\tilde{y}-y\rangle+\langle\tilde{y}-y,\tilde{y}-y\rangle$$
> which shows that $c_x(y)$ is locally semi-concave in $y$ locally uniformly in $x$ (Condition 1). Condition 3 follows from the absolute continuity of $\mu$ (assumed in Section 2). Condition 4 follows from the continuity of $c$ and compactness of $\mathcal{X}\times\mathcal{Y}$, which together ensure that the infimum in the Kantorovich problem is finite. Hence, Conditions 1, 3, and 4 of Theorem A.2 are satisfied.
>
> ---
> > [Q1] Clarification of the superscript “c” in $-v^{*c}(y)$.
>
> **A.** We revised our manuscript to clarify this notation. Here, the superscript $c$ denotes the $c$-transform of $v^*$, defined as $v^{*c}(y)=\underset{x\in \mathcal{X}}{\inf}(c(y,x) - v^{\*}(x))$

---

> > ### Author Rebuttal · Reviewer_yhE9 · 2026-04-02
> >
> > The authors have addressed some concerns, but the main concern about their explanation of the additional $c_q$ and the problems to which the algorithm can be applied remains unconvincing.
> >
> > The inverse problem is a very big area, and the numerical experiments are just some image problems, so a more accurate name would be better because of the additional $c_q$.
> >
> > Though the introduction of $c_q$ improves performance, its motivation and explanation remain unconvincing, especially when the input and output have different dimensions. Even when they have the same dimensions, e.g., difficult image registration, does the term $c_q$ improve performance as well?

---

> > > ### Author Response · Authors · 2026-04-03
> > >
> > > We appreciate the reviewer for the continued engagement and for the opportunity to clarify the remaining concerns.
> > >
> > > ---
> > > **A.** Thank you for your thoughtful comments. We agree that the scope of our method should be described more precisely. In particular, our framework is mainly designed for image inverse problems in which the measurement and target can be compared in the same dimension. To make this scope clear, we revised the manuscript accordingly, added this point as a limitation in the conclusion, and changed the title to ***UOTIP: Unbalanced Optimal Transport Map for Unpaired Image Inverse Problems***.
> > >
> > > Regarding the quadratic cost $c_q (y,x) = ||y-x||^2$, we understand the reviewer’s concern that it may appear to restrict the method to problems where the measurement and target have the same dimension. However, $c_{q}$ can be generalized to $||Q(y) - x||^2$, where $Q$ maps measurements into a space comparable with the target. This is closely related to the setting considered in prior works such as KIDOT [1] and OT-CycleGAN [2]. These works address MRI reconstruction, a representative inverse problem in which the measurements are acquired in $k$-space (i.e., frequency domain), while the reconstruction target lies in the image domain.
> > >
> > > Moreover, even under the theoretically guaranteed regime, we can generalize the quadratic cost $c_q$ to $||Q(y) - x||^2$ if we have a prior information about the task. The theoretical requirement for $c_q$ is the **twist condition** which can be satisfied by cost functions beyond the standard quadratic. For example, when $Q(y)$ is a matrix multiplication and $Q^{T}$ is injective (e.g., $Q$ is a rigid motion), then the twist condition is satisfied. Designing such problem-specific costs based on prior knowledge would be an interesting direction for future work.
> > >
> > > Regarding image registration, we consider it conceptually distinct from an inverse problem. While inverse problems aim to recover a clean signal from measurements generated by a forward operator, image registration focuses on aligning images through spatial transformation.
> > >
> > > [1] Zheng, Taoran, et al. "Towards Prospective Medical Image Reconstruction via Knowledge-Informed Dynamic Optimal Transport." NeurIPS 2025.
> > > [2] Sim, Byeongsu, et al. "Optimal transport driven CycleGAN for unsupervised learning in inverse problems." SIAM Journal on Imaging Sciences.
> > >
> > > ---
> > > We hope these clarifications are helpful. If so, we kindly ask the reviewer to consider re-evaluating the manuscript.

---

### Official Review · Reviewer_UDdC · 2026-03-12

**Soundness:** 3
**Presentation:** 3
**Significance:** 2
**Originality:** 3
**Overall Recommendation:** 4
**Confidence:** 4

**Summary:**

This paper proposes a UOT-based solver, named unbalanced optimal transport map for inverse problems, and showcases its performance on unpaired image inverse problems.  Specifically, it learns an unbalanced optimal transport map from the noisy measurements to the corresponding clean images.  This method can be applied when the noisy and clean datasets have different sample sizes because it does not require paired data.

**Compliance With Llm Reviewing Policy:**

Affirmed.

**Final Justification:**

The paper is technically sound and clearly written, and I appreciate the attempt to extend OT-based inverse problem solvers to the unpaired setting. The rebuttal was helpful in clarifying the role of the quadratic cost and in providing some additional limited-data evidence; in particular, the comparison against DPS with the same 5,500 clean images partially addresses sample efficiency, though it still does not provide a formal or systematic sample-complexity picture. However, my main concern remains unresolved: the paper does not convincingly demonstrate the practical regime in which the unpaired formulation is actually needed. In the experiments, the measurements are synthetically generated from clean images using known forward operators, so paired data are effectively available in the studied settings, even if training is performed in an unpaired manner. The rebuttal points to computationally intensive PDE-based inverse problems as a motivating use case, but this remains hypothetical in the current paper, and it is not yet clear that the proposed training pipeline is practically attractive there, since the method relies on a likelihood-based cost involving repeated evaluations of the forward operator and, in practice, its gradients. As a result, although I find the method technically interesting and the empirical results promising, I do not think the paper currently makes a sufficiently compelling case for the significance of the unpaired formulation. On balance, I view this paper as falling somewhere between weak reject and weak accept. For the record, I choose weak accept, taking into account the authors’ effort in addressing the concerns.

**Key Questions For Authors:**

1. As mentioned in the weaknesses, paired datasets can be constructed for the numerical experiments considered in the paper.  I'm interested to see whether the method can be tested on other numerical examples where paired datasets are difficult to construct or the mismatch is so large that paired datasets are not practical.  For the latter, I'm thinking about, for example, a small set of memory-intensive clean targets (high-resolution images) with a large set of memory-efficient data (hundreds of description of the image in words).

2. The quadratic cost is used in the theoretical analysis for uniqueness of the OT inverse problem, which is then adopted in the application.  However, the term seems to impose structural similarity between the measurement and the target.  I am wondering if that is the reason why the method does not outperform the previous methods by a large margin in the super-resolution task, where the structural similarity lacks the most out of the four tasks.

3. Knowing that he modified cost works for the super-resolution task, I think that it would be helpful to discuss how the quadratic cost can be modified by incorporating the prior knowledge of the forward map A. For instance, if A has rigid transformation, one might consider a cost (||T(x)-y||^2), where T compensates for the transformations.

4. What is the sample complexity of the method, in terms of the number of clean images required, compared with that of other methods?

**Limitations:**

Yes.

**Strengths And Weaknesses:**

Strength:
1. The proposed method is technically sound and is supported by theoretical justification.
2. The proposed method is well motivated in the introduction Section.
3. The paper has a clear structure and polished language..
4. The numerical experiments clearly demonstrate improvements over some of the existing methods.

Weaknesses:
1. It is great that the proposed method accommodates dataset mismatch.  However, for the numerical experiments considered in the paper,  given the known forward map, paired datasets can be easily constructed.
2. The assumptions imposed on the forward map A are rather restrictive, specifically Lipschitz continuity and differentiability.  Furthermore, an additional quadratic cost function is required for the cost function to satisfy the twist condition.

---

> ### Author Rebuttal · Authors · 2026-03-31
>
> We sincerely thank the reviewer for carefully reading our manuscript and providing valuable feedback. We hope our responses to be helpful in addressing the reviewer's concerns.
>
> ---
> > [W1] Paired datasets can be easily constructed for the experimental settings considered in the paper.
>
> **A.** In our setting, only the operator type and noise type are assumed to be known, while the noise intensity, $\sigma_y$, is unknown. Therefore, a paired dataset cannot be directly constructed under our setting. We agree with the reviewer, however, that when the forward operator and noise type are fully known, it is possible to generate **synthetic paired data** by applying $A$ to clean images and adding noise. To investigate this setting further, we conducted additional experiments by augmenting our model with a supervised loss term:
>
> $\mathcal{L}=\mathcal{L}\_{unpaired}+\mathcal{L}\_{paired} \quad \text{where} \quad \mathcal{L}\_{paired} = \|| T_{\theta}(y) - x \||_{2}^{2}$
>
> Across both Gaussian deblurring and super-resolution, UOTIP consistently benefits from the addition of supervised pairs. These results suggest that **UOTIP can effectively incorporate supervised information and further improve when paired data are available**. The results are as follows:
>
> > Gaussian Deblurring
>
> ||Method|PSNR ↑|FID ↓|
> |:--|--|--|--|
> |FFHQ|OTUR|23.82|24.337|
> ||UOTIP (Ours)|_24.06_|_21.210_|
> ||UOTIP (Ours + paired)|**25.03**|**15.829**|
>
> > Super Resolution 4x
>
> ||Method|PSNR ↑|FID ↓|
> |:--|--|--|--|
> |FFHQ|OTUR|24.09|22.751|
> ||UOTIP (Ours)|_24.35_|_19.475_|
> ||UOTIP (Ours + paired)|**25.01**|**12.145**|
>
> ---
> > [W2] & [Q3] Strong assumptions on the forward map $A$ and modification of the quadratic cost by incorporating prior knowledge of $A$.
>
> **A.** The operators considered in our paper satisfy Lipschitz continuity and differentiability. In addition, as discussed in Lines 302-307, the super-resolution setting involves interpolation, which does not satisfy these assumptions, yet our method still shows strong empirical performance in this case. This suggests that the practical applicability of the method is not limited only to operators that strictly satisfy these conditions.
>
> Regarding the quadratic cost, we agree that it is introduced to satisfy the twist condition in the theoretical analysis. However, from a practical perspective, Figure 4(b) and Table 7 show that the method still works well even without the quadratic term, that is, when only the IP term is used. As the reviewer suggests, incorporating prior knowledge of $A$ into the cost design is a promising direction. For instance, when $A$ admits an inverse, a modified cost of the form $\||A^{-1}(x) - y\||_2^2$ may be more suitable. More generally one may consider left- or right-inverses when a full inverse is unavailable. However, any such cost must still satisfy the twist condition to fit within our theoretical framework. We believe that designing problem-specific costs based on prior knowledge of $A$ is an interesting direction for future work.
>
> ---
> > [Q1] Evaluation of the method on examples where paired datasets are difficult or impractical to construct due to severe dataset mismatch.
>
> **A.** As the reviewer suggested, our unsupervised setting would be promising when paired data cannot be constructed or when the mismatch between source and target is so severe that pairing is impractical. Such examples include inverse problems from computationally intensive PDE simulations, where generating paired data is prohibitively expensive, and the cross-model scenario that the reviewer describes (a small set of high-resolution clean images alongside a large collection of text descriptions). In the latter, the source and target domains differ not only in scale but in modality itself. We believe the UOT formulation is well-suited for this setting, since its relaxed marginal constraints naturally accommodate class imbalance and distributional mismatch. While these directions fall outside the scope of the current work, we agree that they would be an important future research direction.
>
> ---
> > [Q2] The quadratic cost as a possible factor limiting the performance gain in super-resolution, where the structural similarity between the measurement and the target is the lowest among the four tasks.
>
> **A.** To examine this point, we evaluated the structural similarity between the measurements and the targets for each task using SSIM on AFHQ.
>
> |Operator|SSIM|
> |--|--|
> |Super Resolution 4x|**0.7159**|
> |Gaussian Deblurring|0.5211|
> |HDR Reconstruction|0.5850 |
> |Nonlinear Deblurring|0.7079|
>
> The results show that the super-resolution task actually exhibits the highest structural similarity among the four tasks, meaning that the gap between the measurements and the targets is smallest in this setting. This suggests that the relatively smaller gain of our method in super-resolution is more likely due to the limited room for improvement, resulting from the common use of the quadratic cost among OT baselines.

---

> > ### Author Rebuttal · Reviewer_UDdC · 2026-04-04
> >
> > The rebuttal is helpful and resolves part of my concerns. The authors clarify the role of the quadratic cost and provide additional evidence regarding the super-resolution setting. However, I still do not think the paper fully demonstrates the practical regime where the unpaired formulation is most needed, and the sample-complexity question remains unanswered. Therefore, while my concerns are reduced, they are not fully resolved.

---

> > > ### Author Response · Authors · 2026-04-04
> > >
> > > We sincerely thank the reviewer for the continued feedback and for the opportunity to clarify the remaining concerns.
> > >
> > > ---
> > > > **Q.** I still do not think the paper fully demonstrates the practical regime where the unpaired formulation is most needed, and the sample-complexity question remains unanswered. Therefore, while my concerns are reduced, they are not fully resolved.
> > >
> > > **A.** Regarding sample complexity, we provide the details of the dataset used in our experiments in Appendix B (e.g., FFHQ: 5,500 clean images for training). For a meaningful comparison, we additionally report results for DPS [1], a diffusion-based baseline whose original version relies on a pre-trained diffusion model trained on 49,000 clean FFHQ training images.
> > >
> > > The results reported below are obtained under exactly the same setting as in our experiments. Specifically, for DPS, we trained the diffusion model using 5,500 clean images. Under this setting, our method achieves superior performance in terms of PSNR, SSIM, and Test-set FID. In contrast, DPS may exhibit a stronger tendency toward memorization, as evidenced by the large gap between Test-set FID and Total-dataset FID. This suggests that DPS may require more clean training data to generalize well. Overall, these results indicate that our method is more advantageous when only a limited number of clean training images are available.
> > >
> > > |FFHQ|Method|PSNR ↑|SSIM ↑|FID ↓ (Test set)|FID ↓ (Total Dataset)
> > > |:--|--|--|--|--|--|
> > > |**Super Resolution 4x**|DPS|17.00|0.4250|78.440|4.573
> > > ||**UOTIP (Ours)**|**24.35**|**0.7371**|**52.273**|19.475
> > > |**Gaussian Deblur**|DPS|18.91|0.4769|73.788|4.341
> > > ||**UOTIP (Ours)**|**24.06**|**0.7139**|**55.869**|21.210
> > > |**HDR Reconstruction**|DPS|19.38|0.5440|79.358|4.886
> > > ||**UOTIP (Ours)**|**26.02**|**0.8642**|**45.823**|20.840
> > > |**Nonlinear blur**|DPS|19.01|0.4938|73.414|4.387
> > > ||**UOTIP (Ours)**|**28.52**|**0.8841**|**36.974**|11.370
> > >
> > > Our unpaired formulation might be suitable for inverse problems where paired datasets are difficult or impractical to construct. One representative example is computationally intensive PDE-based inverse problems, where building paired datasets can be prohibitively expensive because generating each pair may require repeatedly solving the underlying PDEs. While such settings seem well aligned with our formulation, we have not experimentally validated them in this work. We therefore acknowledge this as a limitation and included this point in the manuscript as an important direction for future research.
> > >
> > >
> > > [1] Chung, Hyungjin, et al. "Diffusion Posterior Sampling for General Noisy Inverse Problems." ICLR 2023.
> > >
> > > ---
> > >
> > > We hope these clarifications are helpful. If so, we kindly ask the reviewer to consider re-evaluating the manuscript.

---

### Official Review · Reviewer_sHz2 · 2026-03-12

**Soundness:** 3
**Presentation:** 3
**Significance:** 3
**Originality:** 3
**Overall Recommendation:** 4
**Confidence:** 3

**Summary:**

This work proposes UOTIP, an Unbalanced Optimal Transport for unpaired image restoration. Central to the approach is a combined cost function that pairs a standard likelihood term with a quadratic regularizer. Theoretically, this quadratic addition is shown to guarantee the existence and uniqueness of the transport map. However, this holds only under specific assumptions that the forward operator must be Lipschitz continuous and operate over matching dimensions. The paper acknowledges that these conditions break down in tasks like super-resolution. Still, the method demonstrates solid empirical performance despite the gap between the formal theory and practical implementations.

**Compliance With Llm Reviewing Policy:**

Affirmed.

**Final Justification:**

My main concern was the claim of state-of-the-art performance without comparison to recent methods, which raised concerns about the validity of the evaluation. The authors have addressed this issue in their response by including additional comparisons with recent models. I will therefore increase my score to 4.

**Key Questions For Authors:**

1. Given the paper’s state-of-the-art claim, could the authors clarify why the empirical comparisons are limited and mostly restricted to OT-based baselines, particularly since the most recent comparison appears to be with a 2024 method?

**Limitations:**

yes

**Strengths And Weaknesses:**

1. The mathematical formulation of the UOTIP framework is solid. The authors also explicitly acknowledge that, for the super-resolution task, the twist condition in Prop. 3.1 is no longer fully satisfactory.

2. Narrow Scope of Baselines (Major):
The paper claims SOTA for "unpaired inverse problems", but the current empirical evidence does not seem strong enough to support such a broad statement. In particular, the comparison is made against only two baselines, and the most recent one is from 2024. With such a limited set of reference methods, it is difficult to judge whether the claimed superiority really holds.

3. Ablation Study (Minor):
The ablation results suggest that the contribution of the IP terms is rather limited.

Presentation
The paper is generally well-written.

Significance
Addresses a relevant unpaired inverse problem setting.

Originality
Combines likelihood and quadratic costs in a reasonably novel UOT framework.

---

> ### Author Rebuttal · Authors · 2026-03-31
>
> We sincerely thank the reviewer for carefully reading our manuscript and providing valuable feedback. We hope our responses to be helpful in addressing the reviewer's concerns.
>
> ---
> > [Q1 \& W2] The paper’s state-of-the-art claim for unpaired inverse problems would be better supported with broader empirical comparisons, including more recent and non-OT baselines.
>
> **A**. Thank you for this insightful question. We would like to clarify that our state-of-the-art claim is made specifically with respect to OT-based methods for inverse problems. Following the reviewer’s suggestion, **we conducted additional experiments by comparing our method with DPS [1] and the more recent OT-based method KIDOT [2]**.
>
> DPS is a relevant diffusion-based baseline that can be compared under our setting. By contrast, many other diffusion-based methods, such as DDRM [3] and DAPS [4], require additional knowledge of the measurement noise, in particular the noise intensity $\sigma_y$, which makes a fair direct comparison with our setting inappropriate. KIDOT, on the other hand, is designed for medical image reconstruction tasks such as MRI and CT, and relies on the adjoint of the operator $A$, which restricts it to linear inverse problems. Accordingly, we additionally included comparisons on linear inverse problems.
>
>
> **Super Resolution 4x**
> ||Method|PSNR ↑|SSIM ↑|FID ↓ (Test set)|FID ↓ (Total Dataset)|
> |:--|--|--|--|--|--|
> |FFHQ|DPS|17.00|0.4250|78.440|4.573
> ||KIDOT|22.17|0.5528|200.633|155.541
> ||**UOTIP (Ours)**|**24.35**|**0.7371**|**52.273**|19.475
> |AFHQ|DPS|16.52|0.3488|46.592|2.064
> ||KIDOT|22.61|0.5920|110.157|83.914
> ||**UOTIP (Ours)**|**24.97**|**0.7142**|**34.345**|15.939
>
> **Gaussian Deblurring**
> ||Method|PSNR ↑|SSIM ↑|FID ↓ (Test set)|FID ↓ (Total Dataset)
> |:--|--|--|--|--|--|
> |FFHQ|DPS|18.91|0.4769|73.788|4.341
> ||KIDOT|21.98|0.5361|239.004|190.314
> ||**UOTIP (Ours)**|**24.06**|**0.7139**|**55.869**|21.210
> |AFHQ|DPS|18.58|0.4070|42.945|1.912
> ||KIDOT|22.17|0.5562|138.191|112.478
> ||**UOTIP (Ours)**|**24.22**|**0.6804**|**32.014**|12.566
>
> As shown in the table above, **our method outperforms both DPS and KIDOT across all metrics.** In particular, the large gap between DPS's test-set FID and full-dataset FID suggests memorization. We will incorporate these results into the revised manuscript.
>
> These additional results further support our claim within the scope of OT-based methods for inverse problems, and we agree that comparisons with non-OT methods would further strengthen the empirical evaluation. We will incorporate these additional results into our manuscript.
>
> $ $
>
> [1] Chung, Hyungjin, et al. "Diffusion Posterior Sampling for General Noisy Inverse Problems." The Eleventh International Conference on Learning Representations.
> [2] Zheng, Taoran, et al. "Towards Prospective Medical Image Reconstruction via Knowledge-Informed Dynamic Optimal Transport." The Thirty-ninth Annual Conference on Neural Information Processing Systems.
> [3] Kawar, Bahjat, et al. "Denoising diffusion restoration models." Advances in neural information processing systems 35 (2022): 23593-23606.
> [4] Zhang, Bingliang, et al. "Improving diffusion inverse problem solving with decoupled noise annealing." Proceedings of the Computer Vision and Pattern Recognition Conference. 2025.
>
>
> ---
> > [W3] The ablation results suggest that the contribution of the IP terms is rather limited.
>
> **A.** Thank you for this helpful comment. As shown in Figure 4(b) and Table 7, the combination of the Quad term and the IP term consistently yields better overall performance than the Quad term alone. Although the improvement is modest, it is consistent across the results.

---

> > ### Author Rebuttal · Reviewer_sHz2 · 2026-04-02
> >
> > Thanks for the response. Based on the authors’ clarification, I will increase my score to 4.

---

> > > ### Author Response · Authors · 2026-04-06
> > >
> > > We sincerely thank you for considering our rebuttal and for the positive score update. We are truly grateful for the time and effort you dedicated to providing such valuable feedback. We will ensure that all your insightful suggestions are fully reflected in the final version of our manuscript.

---

### Official Review · Reviewer_L6qj · 2026-03-12

**Soundness:** 3
**Presentation:** 3
**Significance:** 3
**Originality:** 3
**Overall Recommendation:** 4
**Confidence:** 3

**Summary:**

This paper introduces UOTIP, a novel framework for solving unpaired inverse problems using Unbalanced Optimal Transport (UOT). Unlike standard OT-based approaches, which require exact matching between the distributions of noisy measurements and clean signals, UOT relaxes this constraint—making the method more robust to practical challenges such as varying noise levels, class imbalance, and distribution mismatch. The key contributions are threefold: (1) formulating inverse problems as learning a UOT map guided by a likelihood-based cost function derived from the measurement model; (2) providing a theoretical proof that incorporating a quadratic cost term guarantees the existence and uniqueness of the optimal transport map, even for ill-posed problems; and (3) achieving state-of-the-art performance on both linear and nonlinear inverse problem benchmarks (e.g., FFHQ, AFHQ), with demonstrated robustness to multi-level noise, class imbalance, and diverse noise types.

**Compliance With Llm Reviewing Policy:**

Affirmed.

**Final Justification:**

I'll maintain my positive score as the rebuttal addressed my questions.

**Key Questions For Authors:**

The cost intensity requires task-specific tuning, and the optimal values vary considerably across tasks—stronger practical guidance on selecting this parameter would enhance usability.

**Limitations:**

Yes

**Strengths And Weaknesses:**

Strengths:
1) The paper provides rigorous analysis of the twist condition and proves that the quadratic cost ensures well-posedness.
2) It presents the first application of Unbalanced Optimal Transport to unpaired inverse problems, introducing a likelihood-based cost function that connects OT with MAP estimation.
3) The method is extensively evaluated across four tasks, two datasets, and multiple challenging real-world scenarios—including multi-level noise, class imbalance, and noise-type mismatch—consistently outperforming strong baselines such as NOT, OTUR, and RCOT.
Weaknesses:
1) While the focus is on OT-based methods, recent diffusion models such as DPS and DDRM have achieved strong results on inverse problems. A comparison—even acknowledging the differing settings—would help contextualize performance.
2) The cost intensity requires task-specific tuning, stronger practical guidance on selecting this parameter would enhance usability.

---

> ### Author Rebuttal · Authors · 2026-03-31
>
> We sincerely thank the reviewer for carefully reading our manuscript and providing valuable feedback. We hope our responses to be helpful in addressing the reviewer's concerns.
>
> ---
> > [W4] Comparison with recent diffusion-based inverse problem methods such as DPS and DDRM would help better contextualize the empirical performance.
>
> **A.** Thank you for your insightful comments. Our paper compares OT-based methods for inverse problems, and we additionally include DPS [1] to broaden the comparison beyond OT-based methods, as shown in below. Our method shows strong performance in terms of PSNR, SSIM, and test-set FID. Furthermore, DPS appears to suffer from memorization, as reflected in the large gap between its FID on the test set and its FID computed over the full dataset. As for DDRM [2], it requires prior knowledge of the measurement noise intensity, $\sigma_y$, making a fair direct comparison with our method difficult.
>
> |FFHQ|Method|PSNR ↑|SSIM ↑|FID ↓ (Test set)|FID ↓ (Total Dataset)
> |:--|--|--|--|--|--|
> |**Super Resolution 4x**|DPS|17.00|0.4250|78.440|4.573
> ||**UOTIP (Ours)**|**24.35**|**0.7371**|**52.273**|19.475
> |**Gaussian Deblurring**|DPS|18.91|0.4769|73.788|4.341
> ||**UOTIP (Ours)**|**24.06**|**0.7139**|**55.869**|21.210
> |**HDR Reconstruction**|DPS|19.38|0.5440|79.358|4.886
> ||**UOTIP (Ours)**|**26.02**|**0.8642**|**45.823**|20.840
> |**Nonlinear Deblurring**|DPS|19.01|0.4938|73.414|4.387
> ||**UOTIP (Ours)**|**28.52**|**0.8841**|**36.974**|11.370
>
> [1] Chung, Hyungjin, et al. "Diffusion Posterior Sampling for General Noisy Inverse Problems." The Eleventh International Conference on Learning Representations.
>
> [2] Kawar, Bahjat, et al. "Denoising diffusion restoration models." Advances in neural information processing systems 35 (2022): 23593-23606.
>
> ---
> > [Q1 \& W5] The cost intensity requires task-specific tuning, and clearer practical guidance on selecting it would enhance usability.
>
>
> **A.** Thank you for this helpful question. The cost intensity, $\tau$, was not tuned in a task-specific manner in our experiments. Instead, the same value was used across all tasks ($\tau=0.001$). Table 3 and Figure 21 present how performance varies with $\tau$ for each operator.

---

> > ### Author Rebuttal · Reviewer_L6qj · 2026-04-02
> >
> > Thanks for clarifying my questions. I'll maintain my positive score.

---

### Decision · Program_Chairs · 2026-04-30

**Decision:**

Accept (regular)

**Comment:**

The authors addressed the main concerns raised by the reviewers in their rebuttal, which appeased all reviewers that unanimously agreed that the paper should be accepted. At the same time, the reviewers do note that the empirical evaluation retains some weaknesses, and the authors should clearly discuss the limitations of their evaluation, along with including all the experiments conducted for the rebuttal in the camera ready.